# Materiality of Precarious Housing and Its Relationship with Perception in Society: Case Study in Municipality of Pinal de Amoles

Luis Eduardo López Flores [1],*, Manuel Toledano-Ayala [1], Juvenal Rodríguez-Reséndiz [1] and Miguel Ángel Rubio Toledo [2]

[1] Faculty of Engineering, Autonomous University of Queretaro, Santiago de Queretaro 76010, Mexico; toledano@uaq.mx (M.T.-A.); juvenal.rodriguez@uaq.mx (J.R.-R.)
[2] Architecture and Design Research Center, Autonomous University of the State of Mexico, Toluca 50000, Mexico; marubiot@uaemex.mx
* Correspondence: luis.eduardo.lopez@uaq.mx; Tel.: +52-443-2-27-17-92

**Abstract:** This paper shows the importance of using interpretative, hermeneutic methods in research related to precarious housing, taking as a case study the municipality of Pinal de Amoles, Querétaro, Mexico. Firstly, the phenomenon is characterized through the use of natural semantic networks, where a degree of knowledge of society about the phenomenon is appreciated. Secondly, the perception of the population regarding the causes and those responsible for the problem is compiled to make sense of these independent variables deductively. Finally, through the theoretical analysis of precariousness and poverty measurement, it is revealed that the related studies approach the phenomenon from the perspective of materiality and not the person and his or her technical–cognitive capacities. In conclusion, there are gaps between the knowledge of the causes from which the conditions of precariousness originate and the perception of individuals to the conditions of habitability in precarious housing.

**Keywords:** housing; precariousness; habitat; society; hermeneutics



## 1. Introduction

In society, housing represents the sample of a vital situation, a way of being, thinking, and feeling that reflects reality in a context immersed in it. It is, therefore, essential to identify and interpret the defining features that make it up to understanding how the processes that generate it arise.

According to Muñoz [1], housing and home must take a more central role in the scholarship on the right to the city, not simply because of their material significance as a space and place from which urban residents can access resources but also because of home's much broader emotional, imaginative, multiscalar, and collective significance as a foundation from which networks, individual and collective identities, and opportunities originate and are reinforced.

It is for this reason that there are research methods used for the interpretation of the various situations related to housing, where the subject-centered approach can be seen from a qualitative epistemic position. This offers the ability to understand the individual and his or her reality from the inside in order to have the opportunity to face the problem in a complex way and with a more accurate view of reality.

In the context of long-term group stigmatization and overt denigration, as Cretan [2] explains, habitus adapts to the changing dynamics and possibilities of complex urban figurations to highlight the divergent ways in which people inhabit a racialized position beyond a generalized and marginalized "otherness".

It is necessary to specify that precariousness is latent in our society and is perceived by those who make it up as the accumulation of adverse or contrary conditions for human

welfare. Second, the concept of precariousness is used in the labor field. It has been the initial source of analysis related to the subject, as expressed in Standing [3], where the study of the living conditions of workers in the industrial sector reflected the employers' oppression of workers by subjecting them to hostile and risky working conditions.

In addition to the above, the increasingly lower economic remuneration, caused by the decrease in purchasing power in all first-order items, is reflected especially in poor food, education, access to health services, time to enjoy life and rest, and, most importantly, housing.

On the other hand, housing has been observed in the investigations carried out on materiality, distinguishing its characteristic features in the understanding of how the functionality is affected by the physical causes belonging to the state of the housing and the economic and social conditions only.

For this reason, Costil [4] explains that, in general, interventions in precarious housing are difficult to implement due to the complexity of legal tools, the protection of private property, and the costs of these policies.

Precarious housing theories focus on studying the inadequate living conditions in which people live, especially in developing countries and marginalized urban areas. Some of the gaps in knowledge in this field include:

1. Lack of accurate and reliable data: It is often difficult to obtain accurate data on the number of people living in substandard housing and the conditions in which they live because they are often not covered by government registration and statistical systems.
2. Lack of awareness of the natural causes of substandard housing: Although there is a general understanding of the causes of substandard housing, such as poverty and lack of access to essential resources and services, much remains to be learned about the complex interactions between these factors and how they relate to each other to generate poverty and substandard housing.
3. Need for multidisciplinary approaches: Substandard housing is a complex and multi-faceted problem that requires comprehensive solutions that address the lack of decent housing and social, economic, and political factors more in line with the circumstances.
4. Traditional approaches to addressing precarious housing have tended to be paternal-istic and populist, often resulting in solutions that are not sustainable or that do not meet people's real needs. It is, therefore, essential to involve communities and people living in precarious housing in the design and implementation of solutions to ensure that they are effective and more appropriate.

With this in mind, precarious housing is increasingly associated with the lack of essential services, which causes risky conditions for the health of human beings. This is a constant risk for those who cohabit or live their daily life among the poorly achieved materiality caused primarily by extreme poverty. Therefore, in the collective intellect, precarious housing denotes a space built with the most unfavorable conditions to develop daily activities safely.

Thus, Alkire [5] establishes an intuitive approach using two cut-off lines to identify the poor. The first is the traditional poverty line or cut-off line based on specific dimensions to identify whether a person suffers deprivation concerning that dimension. The second marks how extensive the deprivations a person suffers must be to be considered poor.

For this reason, Alkire [5] proposed a multidimensional measure of poverty called the "algebraic multidimensional poverty measure" (MPMA). This measure is based on the idea that poverty is not limited to a lack of income but also includes a lack of access to essential goods and services such as education, health, housing, and drinking water, among others. The MPMA combines several dimensions of poverty into a single index that reflects the severity of poverty in a given community.

On the other hand, it is well known that in Latin America, multiple populations are referents of the phenomenon of social marginalization based on its inhabitants' low quality of life. This is caused by conditions of social inequality resulting from geopolitical, eco-nomic, and social factors, prioritizing the constant and excessive search for consumer goods

and the generation of material and financial wealth ahead of human beings' guarantees and fundamental rights.

Therefore, according to Díaz [6], neoliberal policies reinforce the process of the commodification of goods and services, turning the city into an object and support of business in a double sense, both in the formal market and also, in its extension, in the very informality where the most vulnerable and excluded sectors in the process of capitalist accumulation resolve their lives.

Concerning Mexico, these phenomena are related to the characteristics of the history and development of the peoples and cultures that make up Latin America. These features are complex variables to be studied to understand the life situation of populations subjected to social and economic marginalization, as in various parts of the world.

It is derived from the preceding that a marked social inequality arises, and CEAM [7] states that ultimately, marginalization is a multidimensional and structurally originated phenomenon of the economic production model expressed in the unequal distribution of progress, in the productive structure, and the exclusion of various social groups, both from the process and from the benefits of development.

According to Cretan [8], the literature on rural places often focuses on a difference between perception and reality. In perceptual terms, the countryside is often associated with ideas of immutability and resistance to the types of change that characterize urban areas. Therefore, a tension emerges between the idea of rural traditions and everyday customs as permanent yet vulnerable features of the cultural and literal landscape, and the material reality of rural diversity and change.

Thus, for example, the lack of essential services such as drinking water and sewage disposal, electricity, and essential hygienic conditions that are indispensable in a home causes a high and constant risk of contracting infectious or viral diseases, particularly at present, where the disease caused by the SARS-CoV-2 virus represents the most critical health problem worldwide as it is considered a pandemic by the World Health Organization.

As Kampf [9] explains, person-to-person transmission of SARS-CoV-2 occurs through autoinoculation in mucous membranes (nose, eyes, or mouth) and contact with contaminated inanimate surfaces, which has drawn increasing attention to the need for prompt and preventive human protective measures to prevent the contamination of people.

Likewise, Rodríguez [10] affirms that labor precariousness, which has repercussions on housing conditions within the framework of limited housing policies, explains the massification of the popular sectors' practices of de-commodification of production and consumption. Although these practices have characterized urbanizations, in a context of pandemic and massive unemployment, they put the neoliberal context of commodification of daily life in crisis, even if only partially or temporarily.

In addition, the habitability conditions are not only focused on aspects such as thermal, lighting, and acoustic discomfort, but aspects that involve living conditions, housing materiality, and how these relationships are established, affecting human beings in their interior, their life experience, and in the construction of their reality.

On the other hand, environmental psychology theory studies how the physical environment affects human behavior and perception. In the context of housing, this includes how the layout, size, lighting, color, noise, and other factors of the home affect the health, well-being, and quality of life of the people there.

As explained by Valtierra [11], the development of informal housing on the peripheries of cities is an option, given the demand for a place to live and the lack of comprehensive and strategic planning in development programs that can offer alternatives for access to adequate housing in a higher percentage.

Especially for people with low economic resources, this is a consequence of the generation of irregular settlements that lead to environmental deterioration in the city, primarily due to production methods, lack of infrastructure services, inadequate living conditions, and human activity itself.

In the same way, Waldron [12] affirms that the population suffering from this deprivation can be manipulated at election time because political parties use this condition as an asset in favor of their interests to obtain positions in the government. Therefore, Waldron [12] states that new approaches are required to understand the interplay between intensifying housing precariousness and rising support for populist politics.

Habitability refers to the ability of a dwelling to meet the basic needs of residents, such as safety, comfort, privacy, and access to utilities. Environmental psychology theory is essential in understanding how these factors are related and how they can be improved to create healthier and more sustainable homes.

Indeed, research addressing precarious housing should focus on the relationship between the people who live in it, their living conditions, and their quality of life. They should also focus on the factors that provoke significant changes in those subjected to this circumstance of life through the political, social, and economic system to provide solutions that complement the response capabilities of individuals to the adverse phenomenon.

It is evident that it is important to have an in depth understanding of the behavior and the prevailing perception of the individual who builds his reality based on his capabilities in contrast with the adverse conditions of life. For this reason, it should be analyzed through the interpretative, hermeneutic epistemological posture from which the investigations that reflect social actions that have as a purpose the improvement of the living conditions of the people who live in precarious housing.

A precarious housing study can benefit society in several ways:

1.  Raising awareness: The study can raise public awareness of the precarious housing conditions many people face, which can spur a broader debate about public policies that address this problem.
2.  Identification of needs: The study can help identify the specific needs of the population living in precarious housing. It can help public authorities and non-governmental organizations design more effective programs and policies to address the needs of this population.
3.  Improvements in the quality of life: The study can provide information that will enable public authorities and non-governmental organizations to improve the living conditions of people in precarious housing. For example, it can provide information on the need for housing repairs, improvements in water quality, and eliminating health hazards.
4.  Poverty reduction: Precarious housing is often an indicator of poverty. A study can help identify policies and programs that can reduce poverty and improve the economic security of people living in these conditions.
5.  Job creation: Programs and policies designed to improve precarious housing conditions can create jobs in the construction and housing industries, contributing to economic growth and reducing regional unemployment.

For all of the above reasons, to understand the essence of the phenomenon of precarious housing, it is proposed that this research be carried out in the Mexican state of Querétaro, specifically in the municipality of Pinal de Amoles.

The above is derived from CONEVAL [13], where it can be seen that the municipality is located between canyons, rivers, and waterfalls, and has figures that place it in first place in state poverty, with 84.6% of its population being divided into two groups: 49.7% in moderate poverty and 34.9% in "extreme" poverty, making Pinal de Amoles the municipality of Queretaro with the highest percentage of poverty in 2015.

Specifically, the population of Pinal de Amoles is located 152 km from the capital of the state of Queretaro, with 20,628 inhabitants as of 2015. It represents 1.02% of the population of the state of Queretaro, with 14,757 inhabitants living in poverty, that is, 71.5%, and 23.2% in extreme poverty, in addition to a population in a vulnerable situation with deprivations on the order of 25%, with significant social issues and priority attention zone 1 [13].

In this way, the municipal capital of Pinal de Amoles is established as the case study of the present research to separate the real characterization of precarious housing from its

materiality and the perception of the society represented by the sample presented in the methodology described below.

## 2. Materials and Methods

This research aims to find the semantic weight and meanings through research instruments that allow determining the degree of importance of the systems and subsystems of reality. Similarly, it aims to determine the degree of awareness of the population that is directly or indirectly part of the phenomenon and to specify the responsible actors, the causes and effects that exist in the collective intellect, and the people who live directly in the phenomenon of housing precariousness.

In order to clarify the methodology used, the research scheme is presented, where the methods and instruments designed and their application are established, determining the expected result in each case. It can be seen in Figure 1.

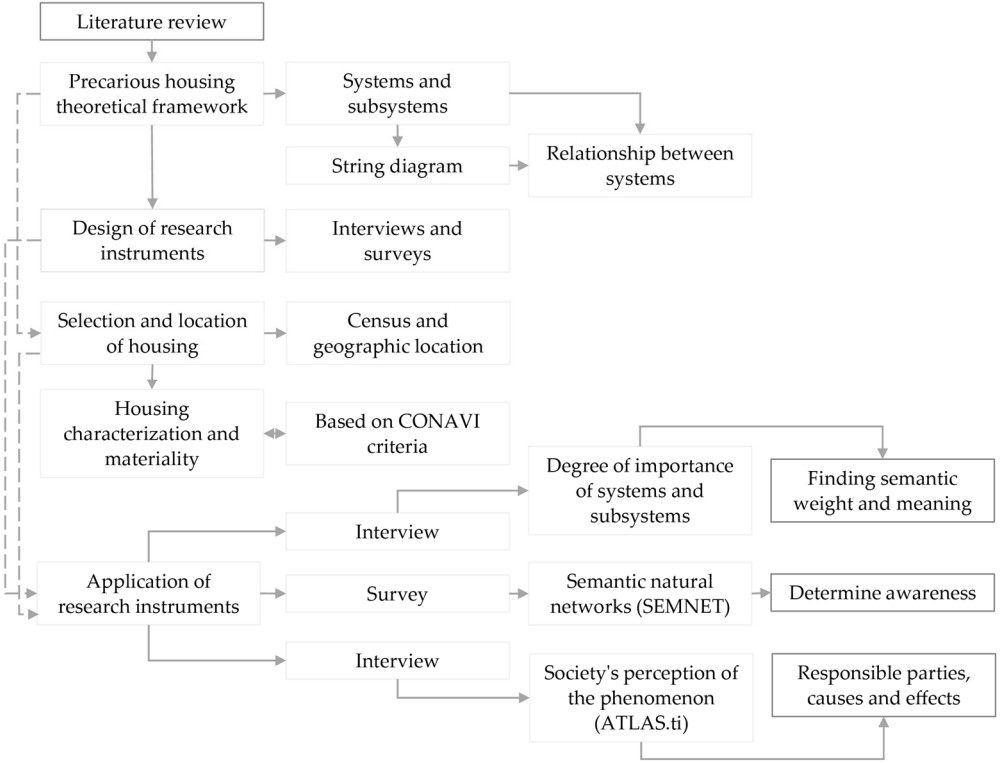

**Figure 1.** Diagram of the research methodology used.

Concerning the methods and materials used for the development of the research, we begin with the definition of the object of study, in line with the analysis of the problem, where it is determined that the municipality of Pinal de Amoles, located in the state of Querétaro, Mexico, is a rural region that reflects the highest levels of extreme poverty according to CONEVAL [14], having the goodwill to allocate a demarcation comprising only the municipal seat to indicate the geographic location of each selected dwelling by way of the census.

In this sense, the projected sample for the reliability of the data obtained is ten dwellings. Care was taken to request the necessary licenses from the municipality authorities for the surveys, interviews, and surveys, as well as the requesting the informed consent of joint agreement from the families that inhabit the houses.

The National Housing Commission, or CONAVI, is a Mexican government institution that provides housing assistance to those who need it most, especially to families living in deprived housing conditions, those with a high index of marginalization or a high index of violence, people with disabilities, and native populations. To do so, this agency establishes

the technical criteria for adequate housing, which serve as a reference for the material state of the homes to be intervened in for applying for habitat improvement programs.

For the selection and exclusion of housing, the technical criteria for adequate housing, formulated by the National Housing Commission CONAVI [15], were used as the indicator of housing quality and spaces.

First of all, the instruments derived from the previous inclusion criteria are the detailed observation of the houses to be considered, as well as the establishment of surveys and interviews that allow us to know the thoughts, feelings, and actions of the population that is living in a house with precarious housing conditions and the society that is part of the phenomenon. Therefore, the objective is to find the relationships between actors, responsible parties, causes, and effects.

This includes two sub-dimensions: the construction material of the dwelling and its spaces. According to these criteria, a population in a situation of a lack of housing quality and space is considered to be those who live in housing with at least one of the following characteristics:

1.　The floors of the house are made of earth.
2.　The roofing material of the house is made of cardboard or scrap.
3.　The walls of the dwelling are made of mud or bajareque, reed, bamboo, or palm, cardboard, metal, or asbestos sheeting or waste material.
4.　The ratio of persons per room (overcrowding) is greater than 2.5.

The next step is to approach the phenomenon of substandard housing from its characterization, which allows the understanding of the material conditions that make it up; a photo gallery of the case studies is made as a result of the previously mentioned reference. In this way, the technical specifications of the materiality of the housing, its physical conditions, the distribution of its interior spaces, and its geographic location are indicated.

In this order of ideas, the design of research instruments is the result of the theoretical framework that allows reality to be studied, from which the subsystems of analysis from which the dimensions are determined are derived. For this reason, the subsystems and the existing relationships to be considered for the design of the research instruments can be seen in Figure 2.

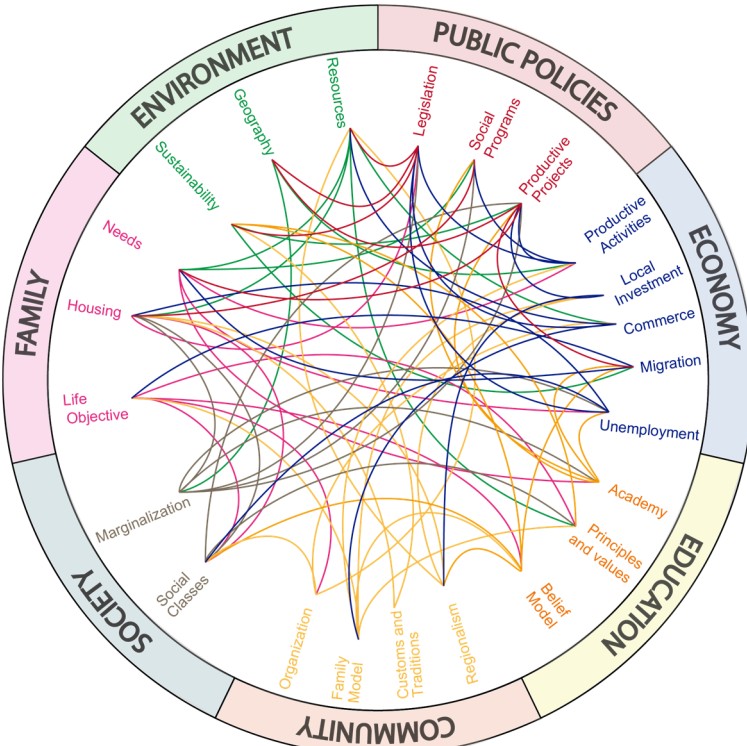

**Figure 2.** String diagram and relationships between subsystems.

Therefore, the questionnaire formats to be applied to the study population within the field research are centered on a questionnaire adapted to the Likert model of a 5-point degree of importance. In this way, five items derived from the scale of values related to the cut of the reality to be observed in 7 dimensions are established:

1.　　Society;
2.　　Family;
3.　　Community;
4.　　Education;
5.　　Economy;
6.　　Environment;
7.　　Public Policies.

In this sense, the instruments will be designed based on closed-ended questions on a Likert scale, with the option of open-ended questions to obtain additional data of possible interest that will be applied to the individual in the family, male or female, who is responsible for and supports the family considered in the census.

The Likert-type scale categorization is a measurement or data collection instrument used in social research to measure attitudes. According to Brunet [16], "it consists of a set of items in the form of statements or judgments among which the reaction (favorable or unfavorable, positive or negative) of individuals is requested".

In turn, to clarify the perception of the society comprising the population of Pinal de Amoles concerning the phenomenon of precarious housing, a survey was designed that includes the seven dimensions previously presented in Figure 2.

Therefore, the research also seeks to determine the actions people take in the face of living circumstances and the conditions of habitability in which they coexist. Therefore, it is understood that substandard housing is not a transversal problem but a longitudinal one, which is affected positively or negatively depending on the attitudes of the individuals involved in the phenomenon.

In this way, three dimensions are added to understand those mentioned: life purpose, technical ability, and technological level.

It is also convenient to resort to the application of a survey that includes the participation of 82 people between the ages of 20 and 47 years old randomly selected in the municipal capital of Pinal de Amoles. This is in response to the need for the surveyed population to belong to the municipal seat where the study of the phenomenon is developed.

In line with the preceding, the present research uses the non-probabilistic stratified sampling method, in which, according to Canales [17], "the cases or units that are available at a given time are taken".

On the other hand, for the processing of the data obtained, SEMNET software will be used to obtain and analyze natural semantic networks (NSN) through the Figueroa technique Figueroa [18]. The technique consists of presenting the subject with a target word that must be defined through nouns, adjectives, and verbs. The subject is given a time limit, which is generally 60 s. Once this time has elapsed, the subject is asked to evaluate how well each of the words he/she wrote defines the target word on a scale of 1 to 10.

The program uses the terminology of the original technique proposed by Figueroa [18]; however, there is also the terminology proposed by Reyes-Lagunes [19], which is equivalent to the original one, where the values calculated by the program are specified as shown in Table 1.

The program includes, in addition to the traditional RSN analyses, the option of performing studies of computational simulations of schemes proposed by Rumelhart [20] and Schvaneveldt's Pathfinder analysis [21]. These methods are a contribution on the part of the program's authors to expand the analysis possibilities of the original technique.

**Table 1.** Comparison between the terminology used in Jesús Figueroa's and Reyes Lagunes' methods. Source: SEMNET (2008).

| Jesus Figueroa | Reyes Lagunes | SemNet |
|:---:|:---:|:---:|
| M-value | Semantic Weight (SW) | Yes |
| Network Richness (NR) | Network Size (NS) | Yes |
| Semantic Distance (SD) | Quantum Semantic Distance (QSD) | Yes |
| SAM Group | Network Core (NC) | Yes |
| Semantic Density (SD) | No equivalent | Yes |
| Q-value | No equivalent | No |
| No equivalent | Effective Load (EL) | No |

In them, different scenarios of concept activation can be simulated, with the idea of analyzing the pattern of interconnection between the different terms.

Based on the above, to reinforce the analysis of the perception and participation of society in the phenomenon, an open interview format was designed to verify and compare the data obtained previously, which reflect the relationship between those who cohabit with cases of precarious housing and those who live in vulnerable conditions as a cause of the problem.

In addition, the ATLAS ti 9 content analysis program will be used for data processing to interpret the texts obtained in the 11 open interviews conducted with people between 35 and 52 years of age located in the region that includes the municipal capital of Pinal de Amoles, in a non-probabilistic manner.

The purpose of the above is to hierarchize, order, and explain the meaning of the arguments given by the participants and to reflect them graphically to facilitate their understanding and interpretation.

Finally, all the above will be compared with the theories of classification, identification, and measurement of extreme poverty, as well as the theories of precariousness and housing in vulnerable conditions to find the contribution to knowledge that will allow the design of innovative methodologies in the area of study.

In this way, the methodology for applying the research instruments and the expected results is established, as shown in Figure 3.

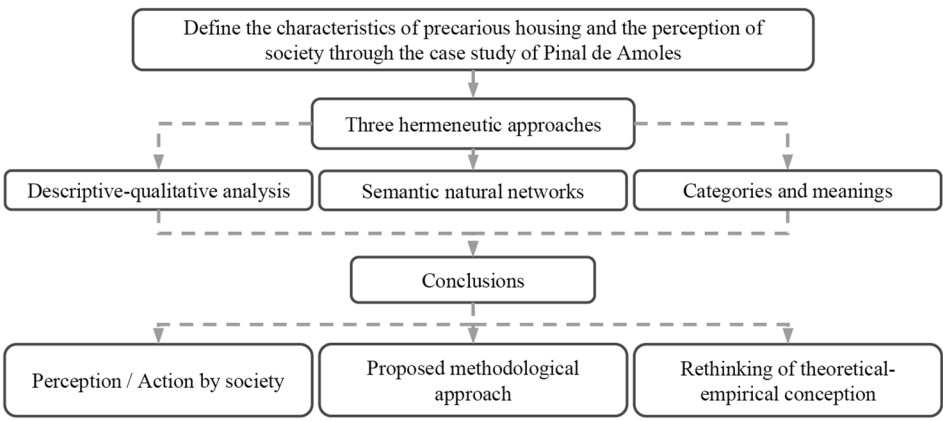

**Figure 3.** Diagram of the methodology used to address the phenomenon.

With this in mind, we proceed to generate a census of houses recognized by their obvious material conditions, which allow their inclusion for the analysis of the state of conservation and the characteristics of a house in a precarious state. In this way, we begin with the geographic location of the ten houses considered on the following map, indicating each one with the symbol (L-x), which refers to its consecutive number, as shown in Figure 4.

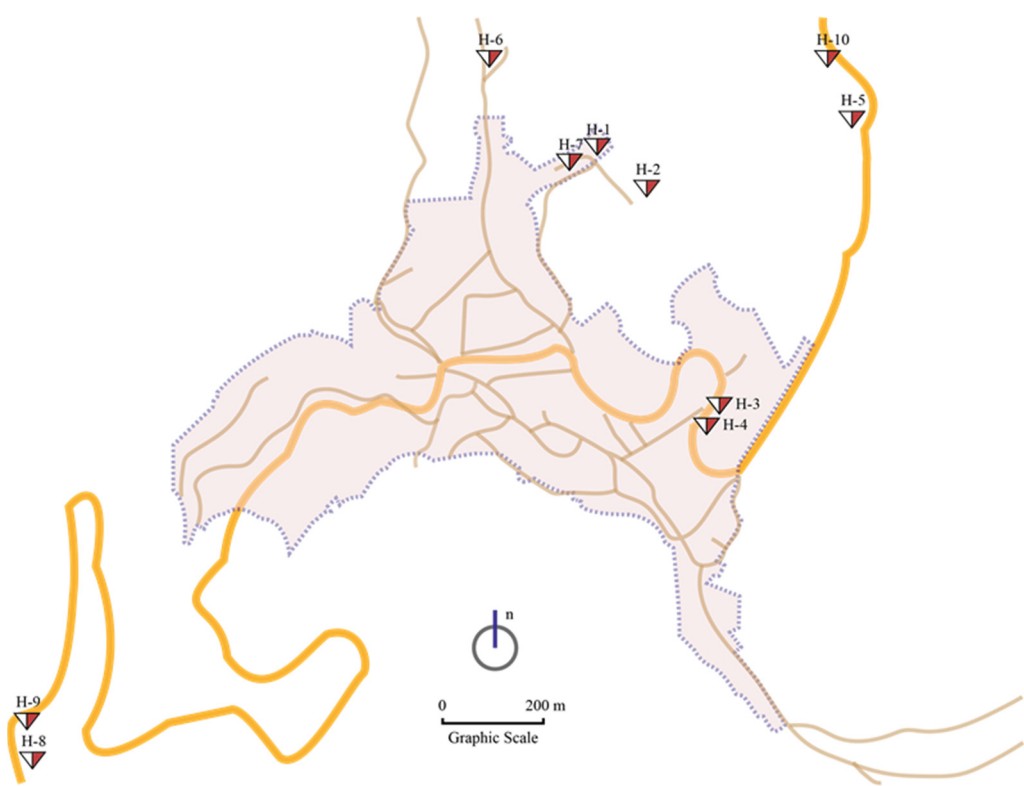

**Figure 4.** The geographic location of the ten houses included in the research.

## 3. Results

### 3.1. Characterization of Precarious Housing

It is necessary to understand that precarious housing is a phenomenon related to the extreme poverty that affects many people worldwide. This type of housing is characterized as a response of people to build their habitat in a poorly achieved way, with the materials that can be obtained from leftovers.

These materials can be several varieties of metal profiles and sheets obtained as waste from construction sites, which have perforations and deformations due to their previous use.

Also used is discarded wood of various types and dimensions with perforations and irregular cuts derived from their previous processes, spliced together to complete the exterior or partition walls. In addition, materials derived from petrochemical processes are used, essentially polymers such as light plastics, tarpaulins, and rigid plastics, used primarily to fit doors and windows.

In the same way, masonry elements based on stone, brick, or adobe made by the inhabitants are used, which sometimes do not have plaster or finishes to counteract the natural effects of temperature and wind inside the houses.

The census is carried out including the necessary data on its excellent location, having a consecutive record of the dwelling, location in geographic coordinates, altitude, built area, and the specific conditions of the materiality of the dwelling, considering the physical conditions of the floors, walls, and roofs. Likewise, the essential services, including energy, sanitation, and access, are identified and analyzed, as shown in Table 2.

**Table 2.** Location and materiality census of the ten houses considered.

| | Site/Legality | | | | | Housing Materiality | | | | | | | | | | | | Basic Services | | | | | |
| | | | | | | | Inner Floor | | | Walls | | | | Roofing | | | | Energy | | | Sanitary | | Access | |
| Housing | Location | Altitude | M² | Owner | Concrete | Earth | Wood | Masonry | Wood | Steel Laminate | Plastic | Concrete | Wood | Steel Laminate | Plastic | Electricity | Ilumination | Gas | Potable Water | Drainage | Sidewalk | Roadway |
|---|---|---|---|---|---|---|---|---|---|---|---|---|---|---|---|---|---|---|---|---|---|---|
| H-1 | 21°8′21″ N 99°37′31″ W | 2370 | 22 | 1 | | 1 | | | 1 | 1 | 1 | | | 1 | 1 | 1 | 1 | | 1 | | 1 | 1 |
| H-2 | 21°8′18″ N 99°37′27″ W | 2360 | 34 | | | 1 | | 1 | 1 | 1 | 1 | | 1 | 1 | 1 | 1 | 1 | | 1 | | 1 | 1 |
| H-3 | 21°8′5″ N 99°37′24″ W | 2310 | 48 | 1 | 1 | | | 1 | 1 | 1 | | 1 | | 1 | | 1 | 1 | 1 | 1 | 1 | 1 | 1 |
| H-4 | 21°8′5″ N 99°37′23″ W | 2305 | 33 | | | 1 | | 1 | 1 | 1 | 1 | | | 1 | | 1 | 1 | | 1 | | 1 | 1 |
| H-5 | 21°8′24″ N 99°37′13″ W | 2280 | 52 | 1 | 1 | | | 1 | 1 | 1 | | | | 1 | | 1 | | 1 | 1 | | | |
| H-6 | 21°8′27″ N 99°37′39″ W | 2380 | 43 | 1 | 1 | | | 1 | | | | | | 1 | | 1 | 1 | | 1 | 1 | | 1 |
| H-7 | 21°8′21″ N 99°37′33″ W | 2370 | 38 | 1 | 1 | | | | 1 | | | | | 1 | | 1 | 1 | 1 | 1 | 1 | 1 | 1 |
| H-8 | 21°7′8″ N 99°39′41″ W | 2570 | 28 | | | | 1 | | 1 | | | | | 1 | | | | | | | | |
| H-9 | 21°7′8″ N 99°39′41″ W | 2580 | 21 | | | 1 | | | 1 | | | | | 1 | | | | | | | | |
| H-10 | 21°2′11″ N 99°33′2″ W | 2330 | 37 | | 1 | | | 1 | 1 | 1 | | | 1 | 1 | | 1 | | | | | | |
| | | | | 5 | 5 | 4 | 1 | 6 | 9 | 6 | 3 | 1 | 2 | 10 | 2 | 8 | 6 | 3 | 7 | 3 | 5 | 6 |

The following is the analysis based on the photo gallery taken of the object of study, which allows the registration of the characteristic features of the houses included in the analysis, where the immediate exterior materiality, their accesses, and interiors are described.

Figure 5 shows the access to the house located in the upper part of the lot, where the floor layout makes it challenging to move around because it does not have tiles that allow for adequate movement, causing a risky condition.

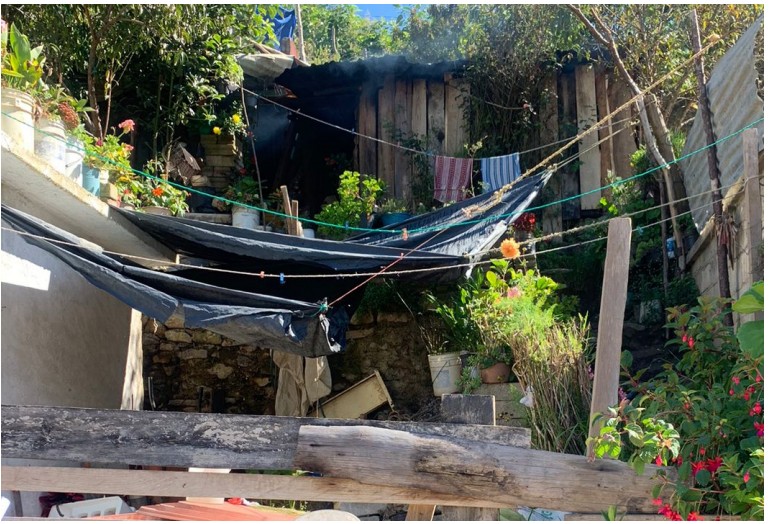

**Figure 5.** Facade and access to the house with front circulation problems.

Likewise, the type of soil is red clay loam, which tends to be pasty and slippery with the excessive humidity of the environment, making it an obstacle even for the inhabitants of the house.

Similarly, the materials used in the house are primarily third-quality pine wood, which does not have any coating or insulation, facilitating its early deterioration.

The walls are laid directly on the ground, transmitting the humidity of the soil into the walls. The interior pavement is made of compressed earth, sometimes with pieces of mortar or concrete without reinforcing steel mesh, completed by small pieces of flagstone.

With regard to the interior spaces, the area used for food preparation has metal sheet stoves, which cause the grease used for cooking food to adhere to the walls and interior

covers, as well as to the scarce furniture, because they do not have extractors, as can be seen in Figure 6.

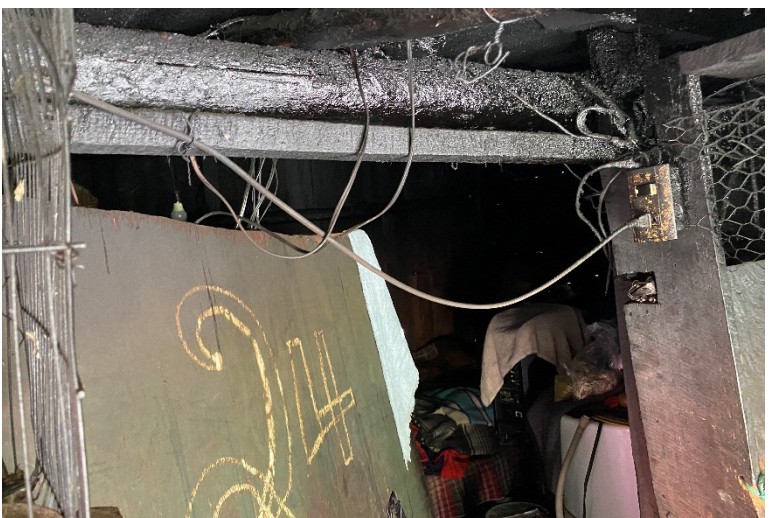

**Figure 6.** Wooden elements inside the house show grease adherence.

Another feature that evidences the functioning of the house refers to the arrangement of the furniture, which may be placed on top of family possessions, which means that there is no adequate space to develop a specific activity, and the space is also affected by the smoke from the cooking area.

Similarly, the cooking utensils are placed on shelves made of wood and stones that also serve as chairs and form an improvised dining area around the stove, as shown in Figure 7.

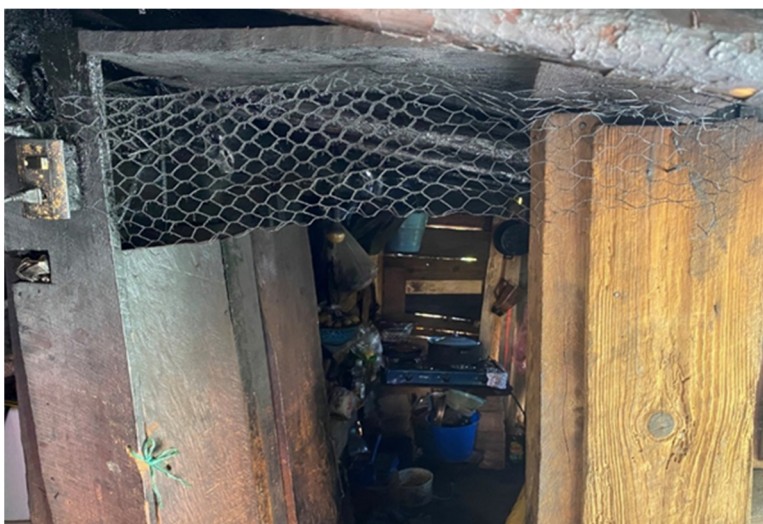

**Figure 7.** Food preparation area and makeshift dining area.

In turn, habitability conditions may be affected by the thermal comfort dimension, affecting the temperature at which activities are carried out inside the house.

This is due to waste or reused metal sheets that are perforated and dented, which does not guarantee the necessary airtightness for thermal control. In addition, no thermal insulation is used, and the sheets are installed without an orderly and logical anchorage, simply trying to confine the desired space, as shown in Figure 8.

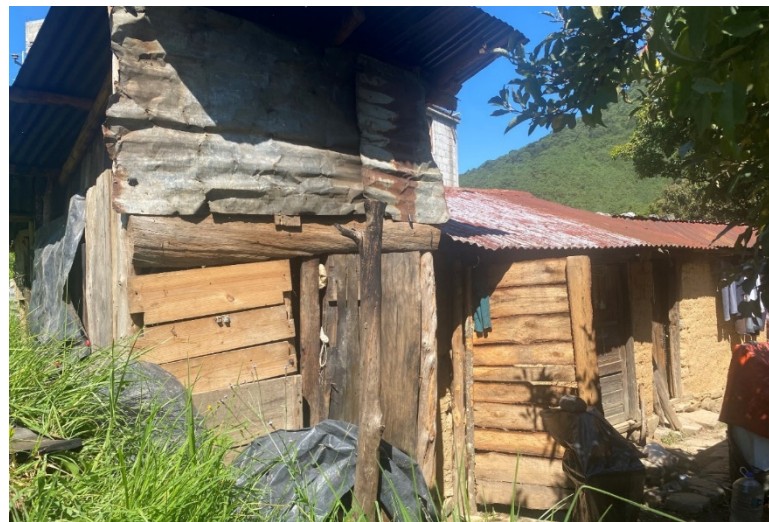

**Figure 8.** Utilization of reused metal sheets with perforations and deformations.

At the same time, the precarious house, being a construction that is achieved with the waste material available, ends up representing the only space that is destined for almost all the activities contained in a house, so there is no arrangement for each space, as shown in Figure 9.

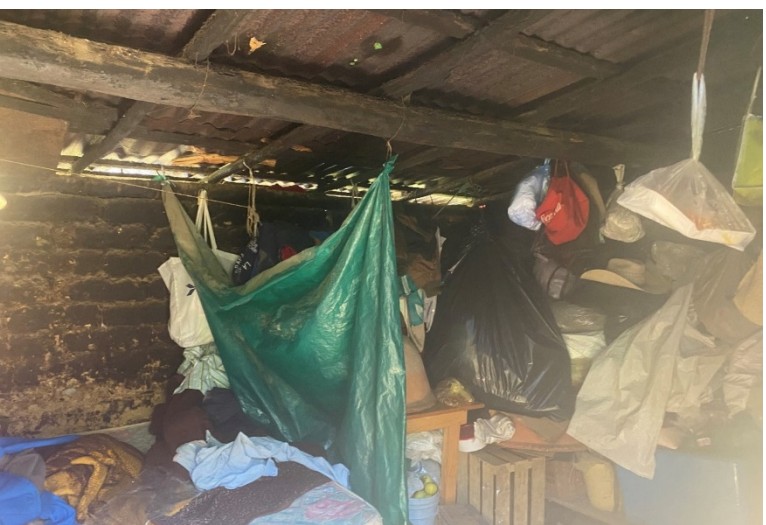

**Figure 9.** Interior space layout of the precarious house.

Something similar happens with the material used to form the roofs, which has perforations resulting from the poor condition of the galvanized steel sheets or asphalted cardboard used to attach to the unprocessed and untreated pine wood structure. With regard to the walls, the wood used can be seen to be matched without achieving an airtight seal, and nails and wires are used to anchor them to the wooden structure made of pine logs, as shown in Figure 10.

Once the house census was conducted, it was established that 50% of the people interviewed were women, and their ages ranged between 35 and 56 years. On the other hand, 50% were men between 38 and 67 years of age. Of the persons responsible for the families, only 40% were owners of the house they live in, but 100% built their houses progressively. Likewise, 80% reported having a coincidentally informal job. Regarding floor area, the houses range from 21 to 52 m$^2$.

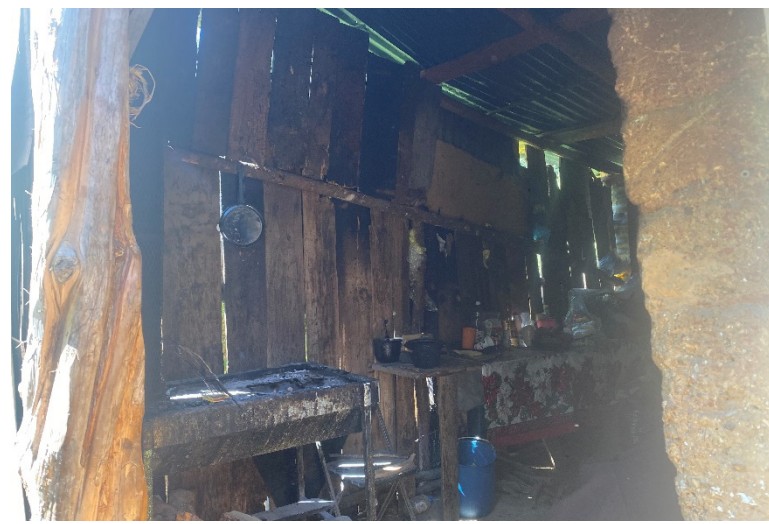

**Figure 10.** Structure, shaping, and anchoring of roof and wall.

In order to determine a parameter for recording the state of conservation of the houses, an index of 1 to 10 was used, where 1 is the lowest quality of conservation of materials and structure, and 10 is the one with the most favorable conditions of conservation. Similarly, an index was used to determine conditions such as safety, accessibility, and thermal and acoustic tightness, as shown in Table 3.

**Table 3.** Materiality census of the houses analyzed for characterization.

| | General | | | Residents | | | | | Condition | | | Status (Index 1–10) | | | | |
|---|---|---|---|---|---|---|---|---|---|---|---|---|---|---|---|---|
| Housing | Gender | Age | Residence Time | Owner | Total | Female | Male | Employment | Self Construction | Built m$^2$ | Number of Spaces | Preservation | Security | Accessibility | Thermal Sealing | Acoustic Tightness |
| H-1 | F | 54 | 20 | N | 7 | 4 | 3 | N | Y | 22 | 3 | 2 | 2 | 1 | 1 | 1 |
| H-2 | F | 35 | 5 | N | 8 | 5 | 3 | N | Y | 34 | 1 | 1 | 2 | 8 | 1 | 1 |
| H-3 | M | 38 | 13 | Y | 6 | 3 | 3 | Y | Y | 48 | 2 | 3 | 3 | 8 | 3 | 3 |
| H-4 | M | 60 | 25 | N | 4 | 2 | 2 | N | Y | 33 | 2 | 2 | 2 | 8 | 4 | 4 |
| H-5 | M | 67 | 42 | Y | 7 | 3 | 4 | Y | Y | 52 | 3 | 4 | 4 | 7 | 5 | 5 |
| H-6 | F | 56 | 21 | Y | 9 | 5 | 4 | N | Y | 43 | 2 | 4 | 5 | 8 | 2 | 3 |
| H-7 | F | 54 | 22 | Y | 5 | 3 | 2 | N | Y | 38 | 2 | 3 | 3 | 1 | 3 | 2 |
| H-8 | M | 45 | 8 | N | 5 | 3 | 2 | N | Y | 28 | 1 | 2 | 2 | 4 | 2 | 1 |
| H-9 | M | 52 | 6 | N | 4 | 2 | 2 | N | Y | 21 | 1 | 2 | 2 | 4 | 2 | 1 |
| H-10 | F | 35 | 6 | N | 2 | 1 | 1 | Y | Y | 37 | 2 | 3 | 3 | 2 | 3 | 2 |

As a result of the above, it is possible to identify cases of overcrowding in houses with up to nine inhabitants in an area of 43 m$^2$, eight inhabitants in an area of 34 m$^2$, and one more with seven inhabitants in 22 m$^2$ of construction. This situation exposes people living in precarious houses to health risks and low quality of life due to the limited space and the lack of privacy to carry out their daily intimate activities.

### 3.2. The Application and Impact of Social Programs and Public Policies on Precarious Housing

In another order of ideas, it is imperative to unveil the extent to which social actions are practical as, theoretically and marginally, social programs are executed from a positivist paradigm as they are susceptible to only quantitative monitoring in the search to reflect

in them the number of resources implemented or managed, in order to record the use of public or private resources to justify them.

Because of the above, social programs focus on something other than the issues that address the complexity of precarious housing since this is approached superficially, keeping the individual out of the deep analysis. The methods used to record and understand the phenomenon need to be made more valid by excluding the formula, the user himself, and the cut of his reality, integrated into the cultural context where the intervention occurs.

In his study, Galiana [22] rightly states that when analyzing the housing problem and contributing to propose solutions to develop appropriate housing policies, making practical the right to decent and adequate housing for the individual and his family as proclaimed in Article 25 of the Universal Declaration of Human Rights (UDHR), among other international norms, should be based on the consideration of four major dimensions, which are interconnected and must be taken into account in order to develop adequate (effective) house policies, namely the political, social, economic, and environmental dimensions.

Thus, any strategy derived from a public policy program focused on improving the quality of life and human habitation must be directly linked to the activities that provide the opportunity to transcend as a person, not only to living in a space with certain family activities.

To explain this fact, Barajas [23] presents an intense review of the historical and budget conditions in the policy to combat poverty in Mexico by the Ministry of Social Development, and points out that in Mexico, social policy is more oriented to the political and electoral issue than to the solution of poverty, or as the author herself puts it, social policy "is not linked to the reduction of poverty in the country, but rather to the need to give viability to an economic model that generates the poor (the programs attend to the people who remain on the margins of development), and to legitimize the government that promotes such a model".

In this sense, Barrera [24] explains how "in Mexico, the most important work was carried out by the federal government through the General Coordination of the National Plan for Depressed Areas and Marginalized Groups (COPLAMAR), which investigated what it considered to be the categories of analysis that explained poverty: food, education, house, health, and income".

However, Sánchez [25] states that the character of these works is to raise the issue of well-being and quality of life from a personal and emotional referent closer to psychology than to other disciplines of the social sciences; therefore, they do not include the actual conditions of people in the analysis.

Therefore, these approaches need a precise contextual framework and consideration of social, cultural, and economic circumstances. They do not propose a scale of well-being and quality of life aimed at observing or capturing the development of functionings and capabilities beyond whether or not the person is satisfied with his or their living conditions. Therefore, their measurements need to be more accurate in describing the actual dimensions and implications of the process.

On the other hand, Canto [26] addresses poverty and inequality in Mexico from the perspective of income. In his work, the author raises, to a large extent, the relationship of poverty with wage precariousness and the inability of some households to acquire a basic food basket.

Thus, in addition to lacking economic resources, this sector of the population needs the tools to jointly apply actions in the field of sustainable development to generate a dignified and stable family economy.

In the same way, the opportunity is wasted to enhance existing skills and empirical knowledge of the use of innovative technologies for human development, economic development, and the improvement of society's quality of life as a whole.

Consequently, inequality plays the most critical role in the lack of opportunities; as Campos [27] determined, "fiscal and economic policy must be changed to combat the inequality in income. In other words, improve taxation, especially for high-income households, and targeted and coordinated spending to reduce inequality and improve the promotion of social mobility".

It is clear that the main discussion between the results of the investigations and the actions analyzed in these leads to a bias in interpreting the information obtained in the field. Therefore, the approach of the recursive strategies must address the main objective, which is the transformation of the habitat and the awareness of what happens to those who experience thinking, doing, and feeling in the precarious housing.

Because of this, the condition of human vulnerability in precarious settlements requires rethinking paradigms, criteria, and phobias in the search for its understanding, so Galiana [22] states that not only the failure to satisfy the right gives rise to social exclusion, since it is not only about protecting the "homeless" but also vulnerable collectives and ensuring non-discrimination, paying particular attention to people living in precarious, unhealthy conditions or urban peripheries without access to supplies ad services.

In addition to the above, an appropriate approach methodology must contemplate a systemic knowledge of the phenomenon's complexity in order to effectively combat the low quality of life and living conditions reflected in the prevailing precarious house, notwithstanding the implementation of social assistance programs in favor of the population.

For this reason, a research instrument comprising an interview and a Likert scale survey is applied to people living in a precarious house, comprising the seven subsystems mentioned previously and the concepts derived from them, to determine the degree of importance of each group of codes mentioned.

As a result of the data obtained, it is possible to state that the community subsystem reflects a lower degree of importance, with 32% in the field of citizen organization, which reflects a non-recurrent lack of interest, but is indicative of the opinion of the member of a house.

To a lesser extent, human development, community organization, general culture, and relating to their neighbors reflect a lack of interest of 45%, followed by public resources and social relations, which indicates a degree of specific lack of interest in what is related to social contribution and community support in the participants. It is possible to observe this in Figure 11.

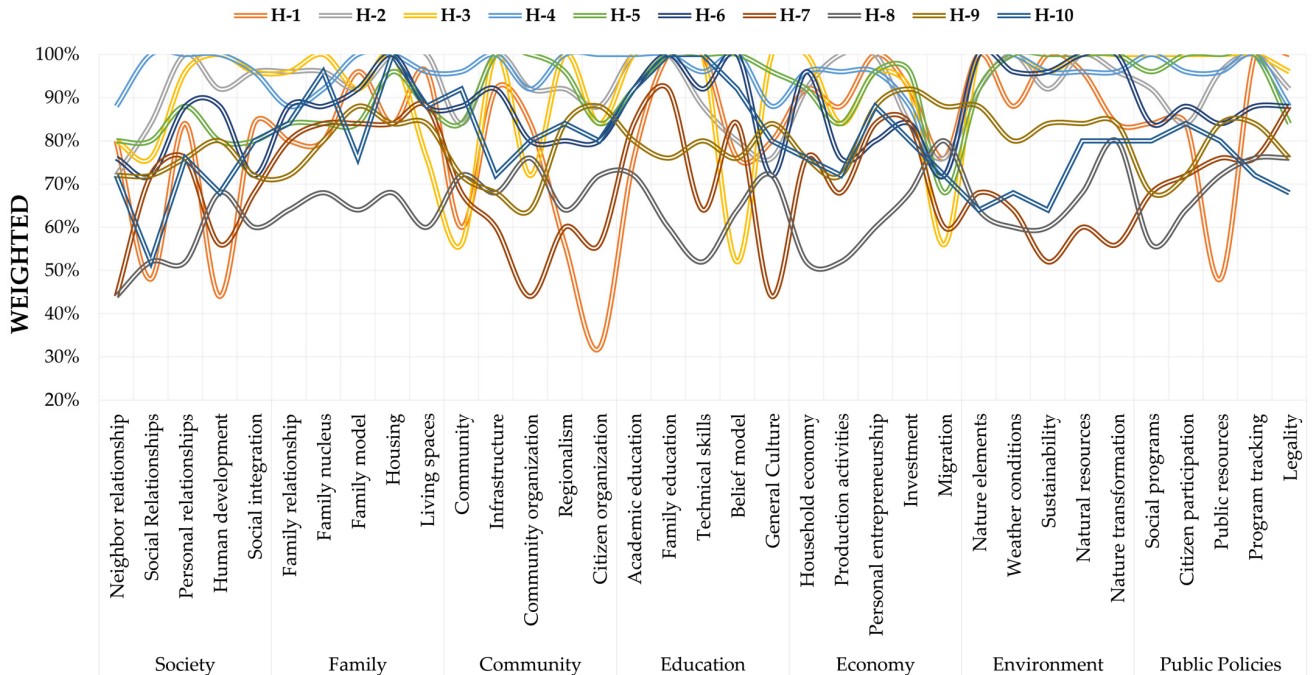

**Figure 11.** Responses were obtained according to the Likert scale of degree of importance.

### 3.3. Natural Semantic Networks as an Interpretation of Reality in the Population

However, to find meaning in investigating the phenomenon, it is essential to gather the perception of the society that cohabits in the population of Pinal de Amoles, especially in the municipal capital.

As a result, the survey was applied in a non-probabilistic manner to collect information from people randomly, covering the perimeter of the municipal capital equitably, as shown in Figure 12.

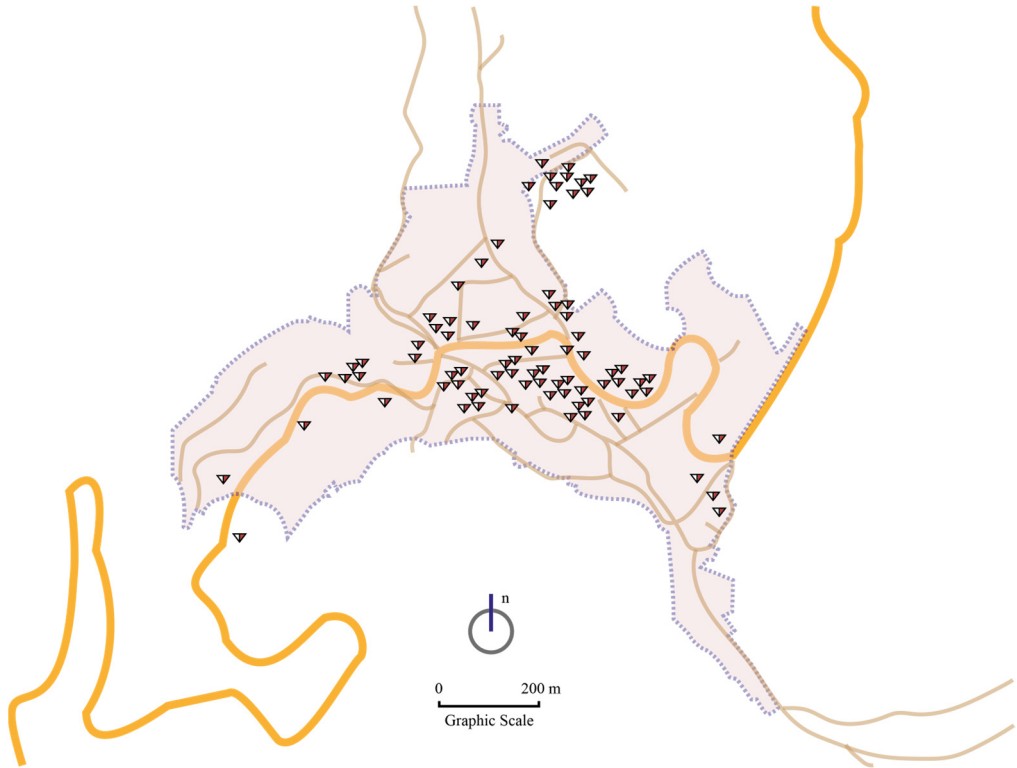

**Figure 12.** The geographical location of natural semantic network surveys.

As mentioned, the research instrument used was determined by a survey of its design where the seven previously mentioned subsystems are contemplated: technical ability, life purpose, and technological level as subsystems of relevance, where the answers obtained are processed utilizing the SEMNET program.

The procedure requires the respondent to observe each subsystem to allow him/her to note the concepts, verbs, or objects that, in his/her opinion, are the most appropriate to the subsystem presented to him/her within 60 s. Once this is complete, a weighting from 1 to 10 of each concept noted is made to obtain a semantic weight, all of which is indicated in Table 4.

As a result, the program allows us to recognize the concepts that are most present in the surveyed population when considering the living conditions and materiality of precarious houses. An example is the subsystem community, where support, union, and security are present when thinking about it. Meanwhile, the economy reflects poverty and money to a greater degree and job to a lesser degree, as seen in Figures 13 and 14.

In addition, the education subsystems refer to support again, but also union and, below that, empathy and security. Likewise, support, union, and love are recognized when discussing the family subsystem, which can be observed in Figures 15 and 16.

**Table 4.** Results of concepts obtained from 82 surveys applied in the region.

| Community | | | Economy | | | Education | | | Family | | | Technical Skill | | |
|---|---|---|---|---|---|---|---|---|---|---|---|---|---|---|
| FB | Definer | FW | FB | Definer | FW | FB | Definer | FW | FB | Definer | FW | FB | Definer | FW |
| 4 | Support | 15 | 1 | Poverty | 25 | 2 | Opportunity | 13 | 4 | Support | 23 | 4 | Learn | 15 |
| 3 | Union | 13 | 2 | Money | 20 | 1 | Illiteracy | 10 | 3 | Union | 16 | 3 | Knowledge | 13 |
| 2 | Security | 10 | 5 | Job | 14 | 2 | Learn | 8 | 1 | Love | 12 | 2 | Job | 10 |
| 1 | Empathy | 9 | 2 | Resource | 11 | 3 | Shortage | 8 | 5 | Job | 7 | 1 | Ingenuity | 9 |
| 2 | Family | 9 | 3 | Shortage | 13 | 1 | School | 9 | 2 | Unit | 6 | 2 | Survival | 9 |
| 2 | Unit | 8 | 1 | Unemployment | 8 | 2 | Basic | 6 | 1 | Sadness | 6 | 2 | Skillful | 8 |
| | NR = 225 | | | NR = 212 | | | NR = 218 | | | NR = 249 | | | NR = 243 | |
| | SD = 6.2 | | | SD = 15.5 | | | SD = 5.2 | | | SD = 12.6 | | | SD = 4 | |
| **Environment** | | | **Technological level** | | | **Public policies** | | | **Life purpose** | | | **Society** | | |
| FB | Definer | FW | FB | Definer | FW | FB | Definer | FW | FB | Definer | FW | FB | Definer | FW |
| 1 | Water | 23 | 1 | Low | 13 | 1 | Corruption | 13 | 1 | Improve | 14 | 1 | Discrimination | 7 |
| 1 | Pollution | 24 | 3 | Shortage | 11 | 5 | Service | 11 | 2 | Job | 15 | 2 | Opportunity | 6 |
| 1 | Tree | 22 | 1 | Advance | 7 | 1 | Government | 10 | 3 | Goal | 12 | 3 | Union | 7 |
| 1 | Air | 16 | 2 | Insufficient | 6 | 1 | Law | 13 | 4 | Overcoming | 10 | 4 | Support | 6 |
| 1 | Nature | 12 | 1 | Null | 7 | 2 | Support | 8 | 2 | Money | 10 | 1 | Rejection | 5 |
| 1 | Earth | 13 | 1 | Development | 6 | 2 | Standard | 6 | 2 | Family | 10 | 1 | Apathy | 5 |
| | NR = 208 | | | NR = 239 | | | NR = 236 | | | NR = 234 | | | NR = 262 | |
| | SD = 11.9 | | | SD = 5.3 | | | SD = 7.5 | | | SD = 7 | | | SD = 2 | |
| **Total items** | | 10 | **Definitions by item** | | | 410 | **Different concepts** | | | 74 | | | | |
| **Total respondents** | | 82 | **Definitions by respondent** | | | 50 | **Common concepts** | | | 17(43) | | | | |
| **Total definers** | | 4104 | **Definitions by respondent by item** | | | 5 | **Open concepts** | | | 57 | | | | |

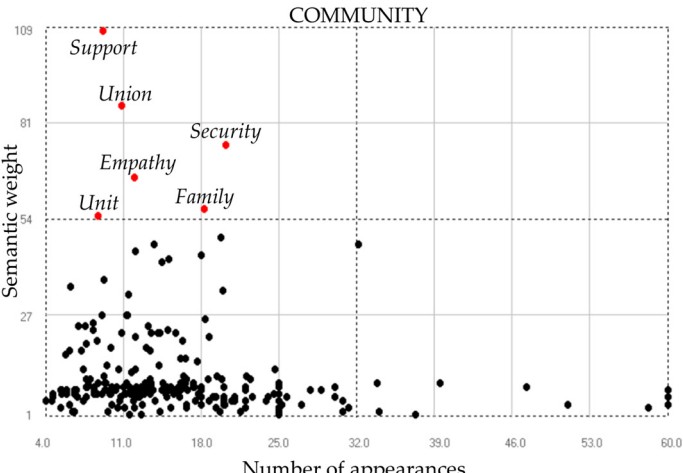

**Figure 13.** Histogram of words obtained for the concepts of community.

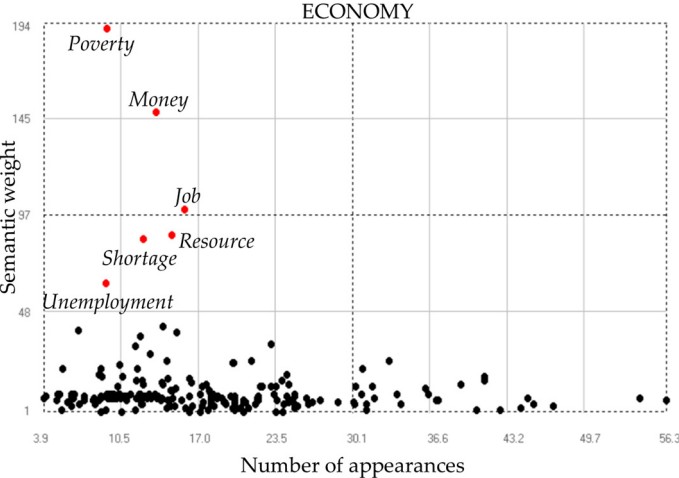

**Figure 14.** Histogram of words obtained for the concepts of economy.

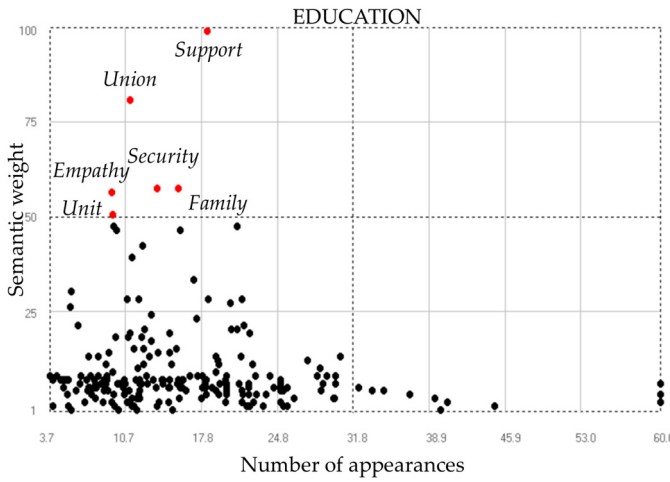

**Figure 15.** Histogram of words obtained for the concepts of education.

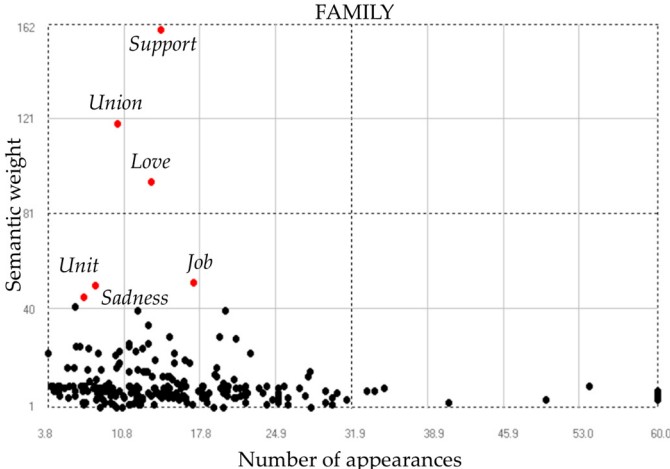

**Figure 16.** Histogram of words obtained for the concepts of family.

In turn, technical ability refers to the concept of learning, knowledge, and work, which are largely congruent. On the other hand, the environment reflects water, pollution, and trees as concepts of concern in the population, as shown in Figures 17 and 18.

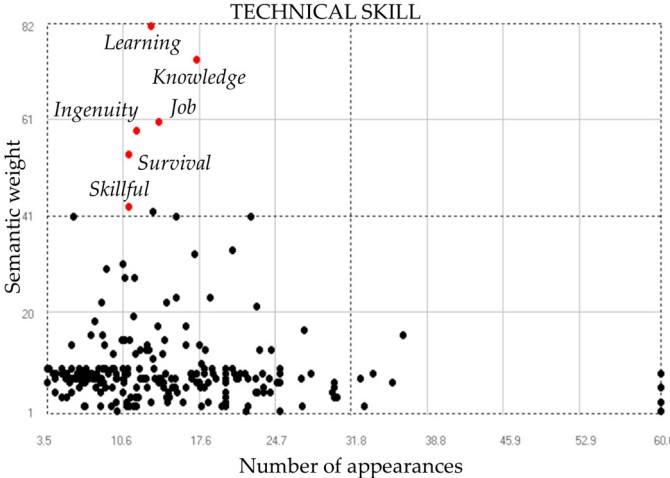

**Figure 17.** Histogram of words obtained for the concepts of technical skill.

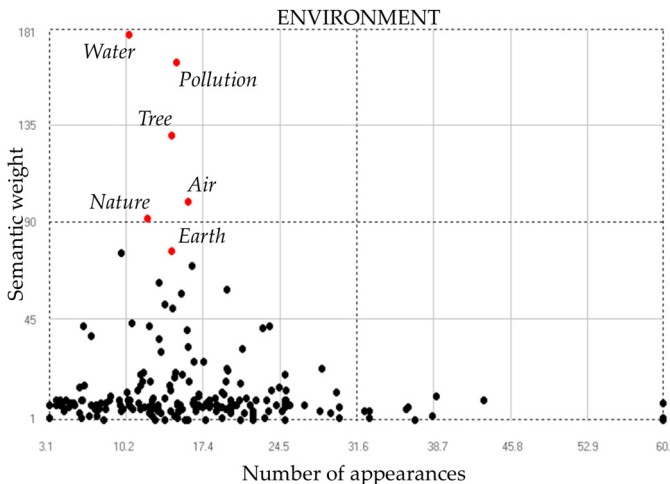

**Figure 18.** Histogram of words obtained for the concepts of environment.

At the same time, the technological level is affected by being considered low and scarce, but it is recognized as a generator of progress in society. Public policies are related to corruption, a topic of great relevance in the respondents' empirical knowledge, but it is recognized that they represent a service that is the government's responsibility as it is responsible for public order, as seen in Figures 19 and 20.

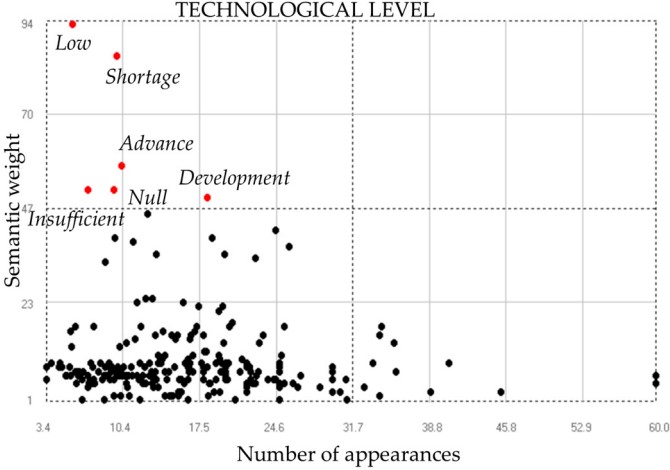

**Figure 19.** Histogram of words obtained for the concepts of technological level.

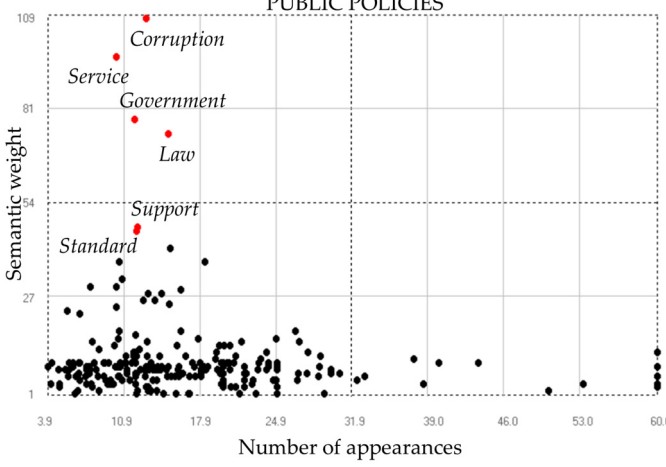

**Figure 20.** Histogram of words obtained for the concepts of public policies.

　　　A similar situation occurs with life purpose, where it can be seen that improvement is a concept that coincides and is congruent with the subsystem, in addition to work and goal as fundamental life objectives. The opposite happens with society, which is contradictorily associated with discrimination as a reflection of it, but opportunity as a trait of hope and support as a primary trait of human accompaniment, which can be seen in Figures 21 and 22.

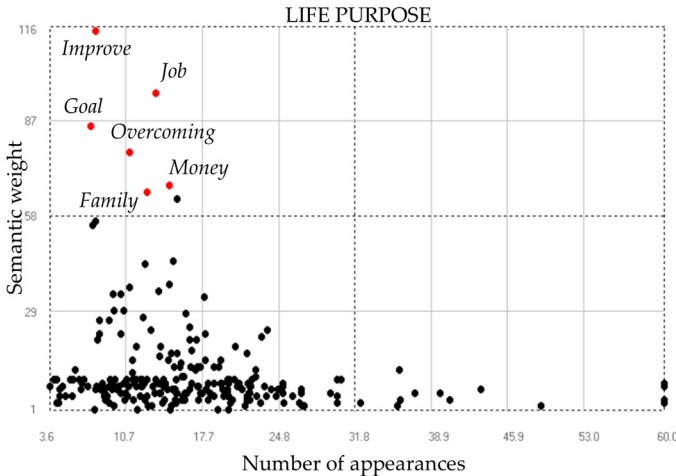

**Figure 21.** Histogram of words obtained for the concepts of life purpose.

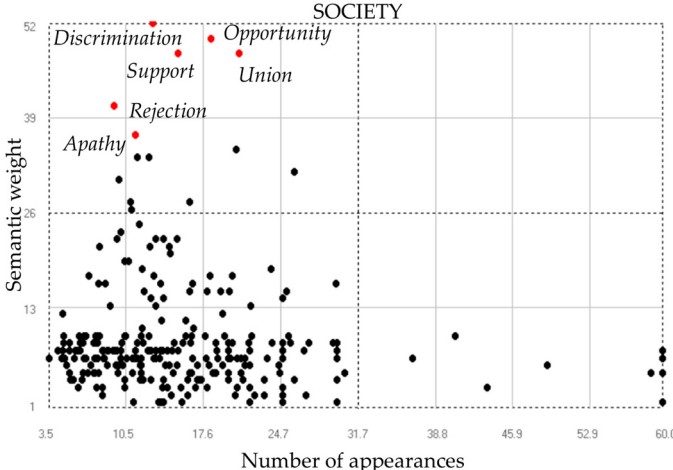

**Figure 22.** Histogram of words obtained for the concepts of society.

　　　In summary, it is possible to determine that the concept of water refers to the primary concern of society regarding the environment and the latent lack of this element of constant and necessary use in the house. Additionally, the common technological understanding confirms the degree of opportunity that exists at the technical level of an individual, which allows continuous improvement and generates a sense of usefulness in society.

　　　Finally, corruption is a recurrent concept regarding the vulnerability of a family living in a precarious house. This is because it is related to public and social assistance policies that may improve families' quality of life, but the opposite seriously deteriorates the social structure.

### 3.4. Society's Perception of Precarious House Living Conditions

　　　It is necessary to emphasize that the perceptions of human beings concerning the same event turn out to be as diverse as the different social constructs. This is because the

individual is subject to the conditions of his environment, reality, and the interpretation of his experiences.

Because of this, the present study aims to discover, through the opinion of the different actors, how the term precarious house is perceived, what degree of knowledge each participant has, and the relationship with his or her environment. The aim is to understand the degree of awareness in society concerning the living situation of people who live in a house with characteristics considered precarious.

The above is achieved through an individual interview in a format of eight questions, which address issues aimed at collecting the feelings and thoughts about the most characteristic features of the issue in question. It also seeks to understand to what extent society is aware of the phenomenon, its participation, and the actions taken to address the problem.

Based on the above, the questions aim to determine the degree of awareness in society about the phenomenon, its participation, and how it affects the society of which it is a part. The questions are the following:

1. What do you understand by a precarious house?
2. What positive and negative characteristics or features define a precarious house?
3. How do you describe the living conditions of the users of a precarious house?
4. What actors do you consider to be involved in precarious housing?
5. What positive and negative aspects do you know about the actions in treating the houses in precarious conditions?
6. What positive or negative effects does precarious housing have on the community?
7. In your opinion, who is responsible for a family living in a precarious house?
8. Do you consider that it is an isolated phenomenon or that it should be addressed? If yes, in what ways?

The inhabitants participating in the interviews comprised seven women between the ages of 31 and 59, with occupations such as being housewives, shopkeepers, and teachers. Additionally, one participant did not study, while two have secondary school studies, one has a high school degree, one has a master's degree, and one has a doctorate.

In addition, there were four male participants, ranging in age from 32 to 73 years old, two of whom worked as teachers, one as a farm laborer, and one in informal commerce.

Finally, the educational level of the male participants is one primary school graduate, one high school graduate, and two more with a master's degree. It is possible to see this in Table 5.

**Table 5.** Census record of 11 interviews and geographic location.

| Interview | Gender | Age | Educational Level | Occupation | Civil Status | Location |
|---|---|---|---|---|---|---|
| I-1 | F | 31 | Bachelor's Degree | Teacher | Free union | 21°08′19.5″ N 99°37′34.1″ W |
| I-2 | F | 32 | High school | Housewife | Free union | 21°08′20.5″ N 99°37′33.7″ W |
| I-3 | M | 73 | Elementary | Merchant | Free union | 21°08′01.8″ N 99°37′52.6″ W |
| I-4 | M | 39 | High school | Journeyman | Married | 21°08′04.7″ N 99°37′31.9″ W |
| I-5 | F | 59 | Not studied | Merchant | Married | 21°08′03.9″ N 99°37′32.1″ W |
| I-6 | F | 53 | High school | Merchant | Married | 21°08′04.0″ N 99°37′32.6″ W |
| I-7 | M | 32 | Master's degree | Professor and researcher | Single | 21°08′06.7″ N 99°37′32.1″ W |
| I-8 | F | 36 | Doctorate | Professor and researcher | Single | 21°08′03.0″ N 99°37′34.3″ W |
| I-9 | F | 38 | Master's degree | Teacher and entrepreneur | Married | 21°08′08.8″ N 99°37′41.7″ W |
| I-10 | F | 33 | High school | Employee | Single | 21°08′08.2″ N 99°37′42.1″ W |
| I-11 | M | 42 | Master's degree | Teacher | Single | 21°08′05.6″ N 99°37′48.6″ W |

With regard to the location where each interview took place, the perimeter of the municipal capital of Pinal de Amoles, Querétaro, was the locality chosen for this research.

It responds to the objective of finding the characteristic features that society associates with the precarious housing phenomenon and which actors, individuals, or organizations may be responsible for the life circumstances of the people who suffer from this problem. In this way, the 11 interviews were placed in a non-probabilistic way, tending to apply the research instruments to people of different social strata and economic activities to compile a broad sample of the proposed topic. The location of each of the interviews, indicated with the symbol (I-x), can be seen in Figure 23.

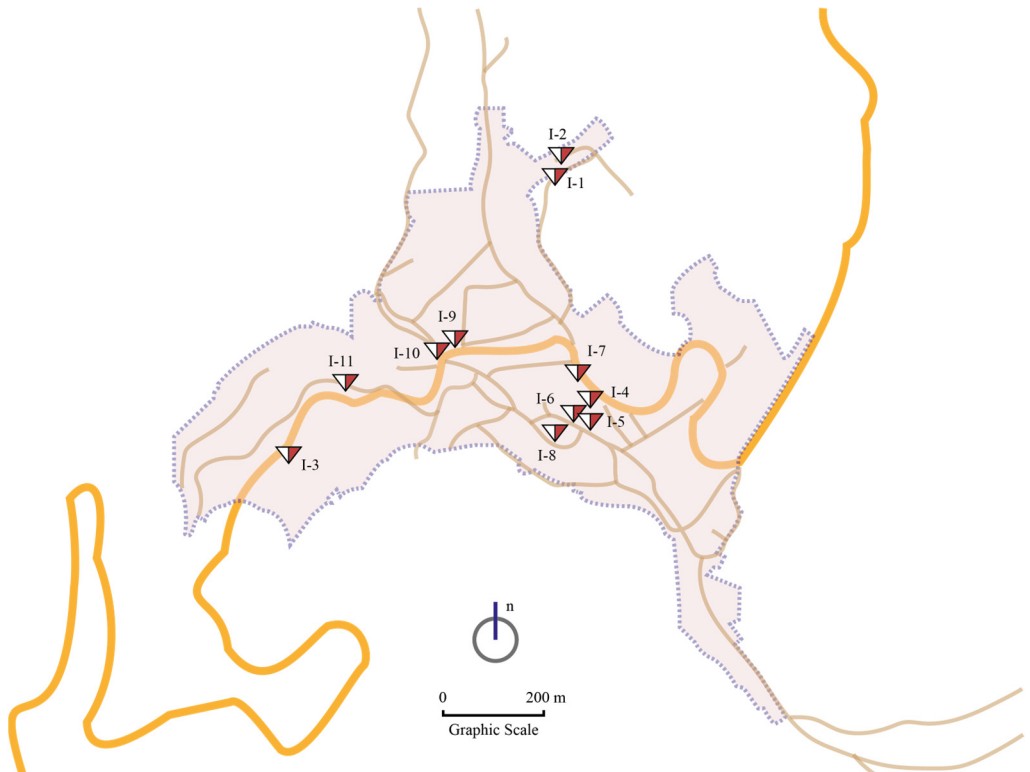

**Figure 23.** The geographical location of societal perception interviews.

The analysis of the data obtained in the field is derived from the above, where the method of inductive textural analysis is used to group the paragraphs in the responses resulting from the interviews, where codes are assigned to define the expressions of the study population.

At the same time, relationships are found between the assigned codes where the meaning of relationship and action is established, where it can be observed that society is aware of the degree of responsibility on the part of government authorities and of the support assigned in public policies.

Furthermore, they recognize that the greatest obstacle to improving people's living conditions in this area is corruption, which is perceived in the different levels of government. In addition, they warn that the lack of knowledge of the problem causes society to not get involved in participatory actions, as this does not affect them directly and may also be the cause of the conformism and negligence of the members of the family living in a precarious house.

Nevertheless, it is found that the lack of job opportunities and the ineffectiveness of social assistance programs significantly impact the low-income population's living conditions. Thus, people's resilient actions in the face of an adverse situation seek to build and maintain the good condition of a house built with waste and garbage, and lacking public services, as shown in Figure 24.

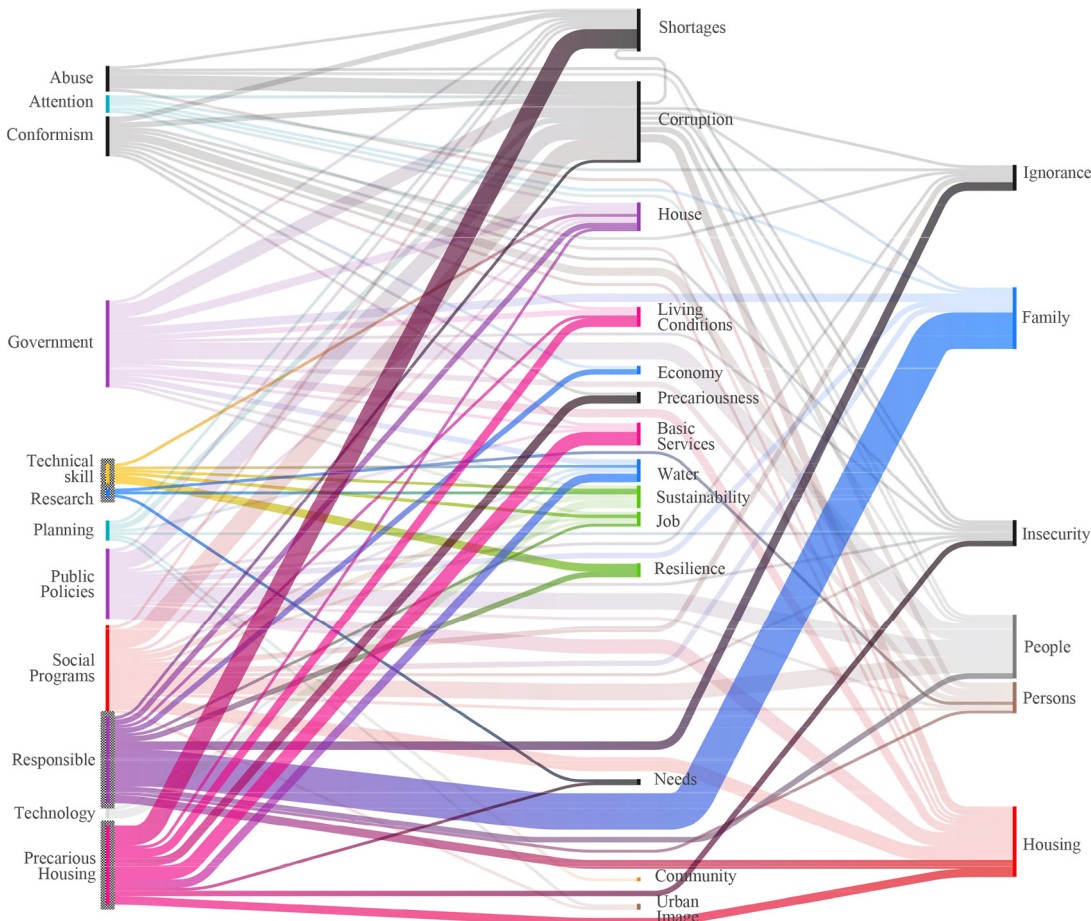

**Figure 24.** Sankey diagram of causes and responsibilities of the phenomenon.

## 4. Discussion

Several theories attempt to explain the conditions of habitability and inequalities in access to a suitable house. Some of them are:

1.  Economic theory: According to this theory, precarious housing results from the lack of supply and demand for affordable houses. When there is a shortage of affordable housing, the price of housing increases, and only people with high incomes can afford to live in adequate housing.

2.  Discrimination theory: Based on this theory, precarious housing results from racial, ethnic, or gender discrimination in the housing market.

3.  Social exclusion theory: According to this theory, precarious housing results from the social exclusion of certain social groups. These groups may include the homeless, refugees, undocumented immigrants, the elderly, or people with disabilities. Social exclusion can result from factors such as lack of employment, lack of access to health and education services, and lack of social support.

4.  Cycle of poverty theory: Based on this theory, precarious housing contributes to poverty and social exclusion. For example, people living in precarious housing may have less access to public services such as education and health care, affecting their health and ability to work and generate income. However, this, in turn, can perpetuate the cycle of poverty.

According to Morales [28], in the face of paradigmatic changes, discourses are sought through which knowledge and praxis obtained through methodologies that do not necessarily come from methods centered on the physical and natural sciences are legitimized; in this sense, rationalities and logics different from those developed in modernity are observed.

From the analysis of the methods presented, the opportunity to strengthen the approach to research aimed at improving the human habitat from the interpretative, hermeneutic posture emerges. Nonetheless, considering that qualitative research is essential to understand the complex problem of the phenomenon, the quantitative aspect is also essential to organize and classify the information obtained from the empirical findings.

The study of precarious housing is of great relevance to the scientific community for several reasons:

1.  It contributes to a better understanding of the living conditions of the people living in these dwellings and the causes and consequences of precarious housing.
2.  It identifies the needs of the population affected by housing precariousness and proposes solutions to improve their quality of life.
3.  It helps design public policies and strategies to address the problem of precarious housing and assess its impact on society.
4.  It fosters interdisciplinary research and collaboration between different fields of study, such as sociology, architecture, engineering, and public health.
5.  It can contribute to developing innovative technologies and materials for safer, healthier, and more sustainable housing.

In summary, the study of precarious housing is relevant to the scientific community because it allows us to understand a significant social problem better and propose solutions that can improve people's quality of life and contribute to society's general development.

At the same time, according to Huchzermeyer [29], there is a need for concerted efforts to include the organizations of the poor in the policy-making process. Socially and poverty-oriented research should support the endeavors of under-resourced community organizations and should support the flow of information within the broader structures of community-based movements to better position them within the policy-making process.

Therefore, it is essential to consider the importance of contemplating an inclusive hybrid paradigm of qualitative methods in the dialectical, phenomenological, or interpretative hermeneutic posture in analyzing the relationships between the subject and his circumstance. Therefore, to find meaning inclusively the individual and his circumstances should be considered at the center of making decisions about his quality of life.

## 5. Conclusions

Living conditions in rural areas of Mexico can be difficult due to the lack of essential services and infrastructure, such as paved roads, electricity, and potable water. In addition, in many rural areas, the economy is agricultural, and people may need help generating sufficient income to meet their essential needs.

With this in mind, Chitonge [30] explains that it is clear that the access of the poor to land and the city can only be protected by political means, in which land destined for the poor is protected from the assault of formal and informal markets.

Nevertheless, precarious housing is a problem affecting many people worldwide, directly affecting people's quality of life and long-term sustainability. It is essential to understand what precarious housing is, its causes, how it affects people's quality of life and sustainability, and, more importantly, who is directly and indirectly responsible for the phenomenon.

In the same sense, it is crucial to understand which strategies can be employed to address the problem of precarious housing. Such strategies include improving housing regulation, increasing access to financing for housing, and providing more resources for improving the livability of affected families.

It is also vital to generate new methodologies for understanding the essence of the phenomenon in order to clarify the circumstances, the actors, and the general conditions in which society directly or indirectly participates.

Precarious housing has been the subject of extensive research, yet many questions still need to be answered.

Future research may be required in the following areas:

1.  Precarious housing causes: Further research is required to understand the root causes of precarious housing. What are the economic, political, and structural factors contributing to precarious housing? What connections do these factors have, and how do they impact various communities?

2.  Impact of precarious housing on health and well-being: Research is needed on how precarious housing affects the physical and mental health and the social and economic well-being of residents. How does precarious housing impact residents' productivity and quality of life? How does precarious housing affect children and youth?

3.  Solutions for precarious housing: Research on effective and sustainable solutions to address precarious housing is needed. What are the best practices for addressing precarious housing? How can government policies and programs address precarious housing? How can communities and non-profit organizations get involved in addressing precarious housing?

4.  Impact of climate change on precarious housing: Research is needed on how climate change will affect precarious housing and how vulnerable communities can prepare to meet these challenges. How will climate change affect the livability of precarious housing? How can communities adapt to and mitigate the impacts of climate change on precarious housing?

In summary, future research may be needed in many areas to address the problem of precarious housing. It is crucial to continue researching effective and sustainable solutions to address this global challenge.

Because of this, precariousness is reflected not only in material conditions but also in people's behavior and the participatory actions of the society of which they are a part. In other words, the complexity of the problem stems from the circumstances of the people themselves but does not depend solely on them, since society allows the inclusion or exclusion of community sectors in unfavorable conditions.

In conclusion, it is impossible to approach the phenomenon of the complexity of the house without generating a method that allows us to understand society's degree of awareness of the problem and how the relationships between society, the agencies in charge of social welfare, and individuals are established.

**Author Contributions:** Conceptualization, all authors; methodology, all authors; validation, all authors; formal analysis, all authors; investigation, all authors; writing—Original draft preparation, all authors; writing—Review and editing, all authors; supervision, all authors; project administration, all authors. All authors have read and agreed to the published version of the manuscript.

**Funding:** This research received no external funding.

**Institutional Review Board Statement:** This research has the authorization of the ethics committee of the faculty of engineering of the Autonomous University of Queretaro under the file: CEAIFI-091-2020-TP, which is ethically approved.

**Informed Consent Statement:** Informed consent was obtained from all subjects involved in the study.

**Data Availability Statement:** Not applicable.

**Conflicts of Interest:** The authors declare no conflict of interest.

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
