# Peer review of "Materiality of Precarious Housing and Its Relationship with Perception in Society: Case Study in Municipality of Pinal de Amoles"

_societies, doi:10.3390/soc13050105_

Round 1

Reviewer 1 Report

The article is rich in terms of results but needs a more solid and clearer structure. Following are some comments and suggestions:

- a knowledge base about materiality is requested

- support the hypothesis presented in the introduction with more references (for example 61 and 64). There are a lot of different arguments that need to be reworked and, maybe, packed. For example: materiality, work effect, psychological impact, etc., which could be linked to the different methods applied and results obtained. 

- please frame in one place the selection criteria of the case.

- the section 2 and the computational simulation need strong work to be unpacked in 3 subsections: methods (the case study and the three methods applied need to be presented in a linear way - ), data collection (hot data were collected and elaborated, which outputs), materials (the case study that needs a deepen analysis and presentation). Figure 1 needs to be contextualised and presented. The methodological scheme, that needs to be put in the first subsection, needs a line regarding results and feedback. Many contents of the section 3 need to be moved to the section 2 (for example 267-282). Figure 3 needs to be put in materials. 

- following sections are really chaotic. there is a meatching of methods, results and materials. For example: 404-423 is a mix regarding materials and literature; 425-435 is a statement for the methods' choice; 436-443 regards methods; 444-447 results; etc.

- please review the discussion and conclusion accordingly to this proposed structure.

- please pay attention to the use of adverbs and the abused use of "above"

- please refine the punctuation and the readability of the text

- please consider the length of the captions and the spaces

Author Response

Reviewer # 1. The article is rich in terms of results but needs a more solid and clearer structure. Following are some comments and suggestions:

Reviewer # 1, Concern # 1: a knowledge base about materiality is requested

Author response: We appreciate your comments to enrich the scientific content of our work. We agree with the concern about the experimental part of this research.

The study proposes a methodological approach that allows characterizing the materiality of the dwellings from the technical capacities of the individuals because, in all cases, it was possible to confirm that they are self-constructed; however, the reference for inclusion or exclusion in the study is based on the technical criteria for adequate housing of CONAVI.

Reviewer # 1, Concern # 2:

support the hypothesis presented in the introduction with more references.

Author response: Thank you, we appreciate your comments.

The arguments presented allow a general understanding; however, it is necessary to reinforce them with more references.

Author action:

The references in the introduction have been expanded to provide better discussion and conclusions.

Reviewer # 1, Concern # 3: please frame in one place the selection criteria of the case.

Author response:

Thank you, we appreciate your comments. We agree that it is a good idea and address this concern in the new version of the manuscript.

Author action:

The selection of the sample members is due to a non-probabilistic selection. Due to the lack of a census provided by the municipality, it was necessary to reference it with the CONAVI criteria.

Reviewer # 1, Concern # 4: the section 2 and the computational simulation need strong work to be unpacked in 3 subsections.

Author response: Thank you very much for your feedback

The new version of the manuscript includes an explanatory methodological chart to resolve the request.

Author action:

A methodological chart explaining the research methods and instruments is included.

Reviewer # 1, Concern # 5: following sections are really chaotic. there is a meatching of methods, results and materials.

Author response:

We appreciate your comments to enrich the scientific content of our work.

Author action:

The sequence of application of instruments in the field allows the presentation of the images and tables according to the steps carried out during the research; that is why the sequence is configured in this way.

Reviewer # 1, Concern # 6: please review the discussion and conclusion accordingly to this proposed structure.

Author response:

We appreciate your comments to enrich the final part of the study.

Author action:

The discussion and conclusion are improved in terms of the necessary references and future lines of research.

Reviewer # 1, Concern # 7: please pay attention to the use of adverbs and the abused use of "above"

Author response: Thank you very much for your observation.

We recognize that there is a preponderant use of the term.

Author action:

A search has been made of using the term within the manuscript, and we have used other terms that allow us to balance it.

Reviewer # 1, Concern # 8: please refine the punctuation and the readability of the text.

Author response:

Thank you very much for your observation.

Author action:

We have reviewed this matter and believe it has been resolved.

Reviewer # 1, Concern # 9: please consider the length of the captions and the spaces.

Author response:

Thank you very much for your observation.

Author action: We consider it to be corrected

Reviewer 2 Report

This article is interesting and could be taken into consideration for publication. The results are good and shed light on the precarious life of marginal people in a Mexican area. However, there are some sections of the paper which should be improved.

First, the introduction should include one paragraph on how this study pushes forward what we know in existing literature of precariousness.

Second, the major problem of this article is it is based on few academic sources (i.e. there are only 21 references in the reference list of the article) and this is a result of poor engagement with current poverty literature both in the literature review/introduction and in the discussion sections. So it is needed to better positioning of this study in the international literature on poverty and precariousness. For instance, authors nicely mentioned in the Discussions that this article is connected to Economic theory, Discrimination theory and Social exclusion theory on which precarious housing is produced but they do not connect their results with similar studies in economic, discrimination and social exclusion theories. For example, in terms of discrimination and social exclusion there are plenty of studies documenting the cases of poor housing of differents groups of Africans (see doi - 10.1016/S0197-3975(00)00037-0, Asians and Latin Americans (doi - 10.7758/RSF.2021.7.2.10), and even the case of the poor Roma people housing and stigma in Europe - see doi 10.1080/1070289X.2021.1920774 and doi 10.1111/1468-2427.13053. On precarity it is also the study of Munoz - doi -10.1080/24694452.2017.1392284. Fleeing from poverty and connected stigma issues against poor migrants were reflected also in other studies - see Wang, 2000 in journal Housing Studies, see O Brien T. et al, 2022 in journal Identities. Moreover, where authors talk about the impact of COVID-19 it has to be mentioned that political parties could use poor people and (their) sensitive environmental background for populism and electoral gains (see Waldron Richard - doi - 10.1177/0308518X211022363, see Doiciar et al, 2021 in Geographica Pannonica, and so on). So more exaples like the above ones have to  be added in the literature review and the discussions of the paper.

Results are nicely presented, but limitations of the data and method have to be added.

Finally, conclusions have to include the international implications of this study or how the outcomes of this study bring new elements to what we currently know in the worlwide housing literature.

Author Response

We would like to thank the reviewers for their efforts and interest in our work. Your comments will improve the quality of this manuscript and give us the opportunity to clarify these comments. We will respond to all the comments raised by the reviewers.

We have carefully reviewed and attended to all comments raised by the reviewers, as listed below, highlighted in the manuscript.

Reviewer # 2. This article is interesting and could be taken into consideration for publication. The results are good and shed light on the precarious life of marginal people in a Mexican area. However, there are some sections of the paper which should be improved.

Reviewer # 2, Concern # 1: First, the introduction should include one paragraph on how this study pushes forward what we know in existing literature of precariousness.

Author response: We appreciate your comments to enrich the scientific content of our work.

Author action: We have included a compendium of knowledge gaps related to the topic and why research is essential.

Reviewer # 2, Concern # 2: Second, the major problem of this article is it is based on few academic sources

Author response: Thank you, we appreciate your comments. We agree with your concern and have addressed this issue.

Author action: We have integrated the references he proposes and other sources that address issues related to neoliberal situations and populism in governmental decisions on precarious housing issues. The above also support the discussion and conclusions.

Reviewer # 2, Concern # 3: Results are nicely presented, but limitations of the data and method have to be added.

Author response: Thank you, we appreciate your comments. We agree with your concern and in the new version of the manuscript we used tables to better present and organize the information.

Author action: We have included a methodological chart explaining the research methods and instruments.

Reviewer # 2, Concern # 4: Finally, conclusions have to include the international implications of this study or how the outcomes of this study bring new elements to what we currently know in the worlwide housing literature.

Author response: Thank you very much for your feedback. We agree with your concern and have addressed this issue.

Author action: The conclusions have been reinforced with the study's implications in various areas where it is possible to interpolate the results and future lines of research.

Reviewer 3 Report

This article provides a clear overview of the research conducted and its findings related to precarious housing in the municipality of Pinal de Amoles, Querétaro, Mexico. It highlights the importance of using interpretative and hermeneutic methods in research related to precarious housing and characterizes the phenomenon through the use of natural semantic networks to evaluate the knowledge of society about it. The article also presents the perception of the population regarding the causes and those responsible for the problem, which is analyzed through deductive reasoning. Furthermore, the theoretical analysis of precariousness and poverty measurement is used to reveal the gaps in knowledge between the causes that lead to precarious housing conditions and the response of individuals to these conditions. The article is well-structured, However, it could benefit from providing more specific details about the findings, particularly regarding the gaps in knowledge that were identified. Overall, this article is informative and provides a good overview of the research conducted. The following suggestion should be considered by the authors in other to increase the internal and external validity of the article:

-The article lacks a variety of literature in the stated field to support the study. There should be a justifiable discussion in the literature part of the study to develop the interrelation. The reason is that I still didn't get the main gap in the study. The relevance of the research problem for the discipline should be highlighted. Why this research is important and how different stakeholders can get benefit from the findings of this study need to be highlighted.   

-Contribution to academia needs to be highlighted in the abstract, introduction and conclusion part of the study. The contribution of the study needs to be explained in such a way that to increase the originality of the study.

- Abstract should cover Introduction and Reason for conducting the research, the Problem (knowledge gap), Methods, Outcomes (results), and Ramifications (Implications).

- Introduction doesn’t have any scientific structure to highlight the problem of the study or the gap in the literature. The introduction of the manuscript is not well-organized author may use the strategy of “ big umbrella” to focus on the main problem of the manuscript.  

- This study needs to be enriched by cited up-to-date references relevant to the research.

- It would be great to work a little bit more on the methodology part to be easily understandable for the readers. Maybe a graphical presentation will help readers to follow the research easily. You may refer to qualitative, systematic reviews or any other methods that are suitable for the research, so it needs some explanation, consider mentioning methods techniques and tactics of the research under a big umbrella.

- There is no coherency in the written text. The authors may need to use some connecting words to increase the fluency of the text.

- Restate the research problem addressed in the manuscript.

- There is not enough evidence and literature to support the LITERATURE REVIEW. I didn’t get what is the gap in the literature and what are the main problems resulting from that. which the author would like to focus on it.

- The conclusion needs to restructure, some essential information which supposes to be in the conclusion part is missing. For example, what are the findings to support the hypothesis of the study? how the author(s) described the contribution of their study to the existing literature? etc., the Conclusion of the study could be much more descriptive in the findings that the author (s) mentioned in the discussion part.

-Suggestion for future study is also missing from the last line of the conclusion. It should be used to point out any important shortcomings of the manuscript, which could be addressed by further research or to indicate directions for further work could take.

Author Response

We would like to thank the reviewers for their efforts and interest in our work. Your comments will improve the quality of this manuscript and give us the opportunity to clarify these comments. We will respond to all the comments raised by the reviewers.

We have carefully reviewed and attended to all comments raised by the reviewers, as listed below, highlighted in the manuscript.

Reviewer # 3. This article provides a clear overview of the research conducted and its findings related to precarious housing in the municipality of Pinal de Amoles, Querétaro, Mexico.

Reviewer # 3, Concern # 1: The article lacks a variety of literature in the stated field to support the study.

Author response: We appreciate your comments to enrich the scientific content of our work.

Author action: We have integrated the references kindly proposed and other sources that address issues related to neoliberal situations and populism in governmental decisions on precarious housing issues. The above also support the discussion and conclusions.

Reviewer # 3, Concern # 2: Contribution to academia needs to be highlighted in the abstract, introduction and conclusion part of the study. 

Author response: Thank you very much for your feedback. We agree with your concern and have addressed this issue.

Author action: The implications of the study and its contributions to academia have been included in the introduction, discussion, and conclusions.

Reviewer # 3, Concern # 3: please refine the punctuation and the readability of the text.

Author response: Thank you very much for your observation.

Author action: These fields have been reinforced due to the compendium of knowledge gaps the study problem poses.

Reviewer # 3, Concern # 4: This study needs to be enriched by cited up-to-date references relevant to the research.

Author response: Thank you, we appreciate your comments.

Author action: Within the knowledge gap, the problems and future research lines have been highlighted to allow the reader to understand the reasons for the study, the scope, the methodology, and the results obtained.

Round 2

Reviewer 1 Report

This second round of review has improved the article but not totally. There are already many parts chaotic and not flowing. 

There are some suggestions to complete this round of review:

- INTRODUCTION: Now the text is too long and not flowing. There are some added parts that interrupt the thread of the speech (for example 113-117) and others that are unconnected (125-150). The part 49-52 needs a solid reference. In some cases, there are too many paragraphs (for example 121, 124). Concern #3 has not been resolved because the text 189-191 is not exhaustive to clarify the selection criteria. Here in the INTRO, criteria need to be briefly introduced and clarify and then in the MATERIALS section need to be deepen. For selection criteria, I intended the reasons you decided to study a determined country, region, municipality and its geopolitical contexts. I suggest eliminating the use of "According to" (197) and leave only the reference.

- MATERIALS AND METHODS: I would suggest to not use "used in general" (209), but comprehensively for example. And I would avoid saying that "the research carried out in this article" (210), for example, the research could be illustrated in the article. The Figure 1 is really useful but needs to be introduced and explained. In this section, I didn't find explanation of the CONAVI and think it is important for the reader to understand what it is. By and large, in this section the reader should read reasons why the research objectives are pursued through using a determined method/s and how. With regards to materials, contents are really poor for readers that not know the selected country and case. Figure 2 could be interesting but needs to be explained as well as Figure 3. 269 needs a number page. Check spacing 286-297. Why the "computational simulation" is a new section? 

- RESULTS: there is already confusion on materials and results. Please check. Figure 4 could be more useful in the section 2.

- DISCUSSION AND CONCLUSION: please use the same order criteria to reframe the texts.

- check editing of bullet points, typos, spacing, and repetitions, and use adequate adverbs

- I would suggest to use always numbers for headings and sub headings to facilitate the work of the reader and give always a reference to the main section (3; 3.1; 3.1.1).

Author Response

Reply to the reviewer 1 comments

We would like to thank the reviewers for their efforts and interest in our work. Your comments will improve the quality of this manuscript and give us the opportunity to clarify these comments. We will respond to all the comments raised by the reviewers.

We have carefully reviewed and attended to all comments raised by the reviewers, as listed below, highlighted in the manuscript.

Reviewer # 1. This second round of review has improved the article but not totally. There are already many parts chaotic and not flowing. 

Reviewer # 1, Concern # 1: There are some added parts that interrupt the thread of the speech (for example 113-117) and others that are unconnected (125-150).

Author response: We welcome your feedback.

Author action: The introduction explains the importance of housing in society, followed by the general conditions of precariousness, and continues with possible causes and prevailing situations.

We focus on Latin America to identify the situation in Mexico. Subsequently, the place where the case study is investigated is established.

For this reason, a rural community was chosen for the study (111-116).

Derived from this, the consequences and the prevailing conditions of precariousness in housing provoke psychological and emotional consequences.

Reviewer # 1, Concern # 2: The part 49-52 needs a solid reference. In some cases, there are too many paragraphs (for example 121, 124).

Author response: Thank you very much. This is an important observation.

Author action: As in the previous case, we tried to explain the situation comprehensively. We address the issue in a complex way, including the aspects that seem relevant and interrelated to us. Although they may seem to be isolated ideas, they are all intertwined and allow us to describe the possible causes.

Reviewer # 1, Concern # 3: Concern #3 has not been resolved because the text 189-191 is not exhaustive to clarify the selection criteria. Here in the INTRO, criteria need to be briefly introduced and clarify and then in the MATERIALS section need to be deepen. For selection criteria, I intended the reasons you decided to study a determined country, region, municipality and its geopolitical contexts.

Author response: We appreciate your interest in clarifying this point.

Author action: Similar to concern #8, Pinal de Amoles is the municipality with the highest degree of poverty in the state of Queretaro. For this reason, the location of this case study was determined. This is explained in the introduction, leaving an explanatory paragraph.

Reviewer # 1, Concern # 4: I suggest eliminating the use of "According to" (197) and leave only the reference.

Author response: We find your suggestion very appropriate.

Author action: The meaning of the reference has been corrected by deleting "in agreement" and leaving the original reference.

Reviewer # 1, Concern # 5: I would suggest to not use "used in general" (209), but comprehensively for example. And I would avoid saying that "the research carried out in this article" (210), for example, the research could be illustrated in the article.

Author response: Your observation is very reasonable. Thank you very much!

Author action: The paragraph has been reworded in response to your suggestion.

Reviewer # 1, Concern # 6: The Figure 1 is really useful but needs to be introduced and explained.

Author response: It is essential to introduce this figure properly. Thank you very much!

Author action: We have included a paragraph that establishes the objectives of the research, which determine the scopes explained in Figure 1 as an introduction.

Reviewer # 1, Concern # 7: In this section, I didn't find explanation of the CONAVI and think it is important for the reader to understand what it is

Author response: This is an important aspect to introduce to the reader. Thank you!

Author action: We have included a paragraph describing the agency, its functions, and its scope as an instrument for the housing selection criteria.

Reviewer # 1, Concern # 8: By and large, in this section the reader should read reasons why the research objectives are pursued through using a determined method/s and how. With regards to materials, contents are really poor for readers that not know the selected country and case

Author response: Thank you very much for your comment.

Author action: We have included a paragraph that clearly explains the objective and purpose of the instruments included to determine causes, effects, and those responsible for the phenomenon according to society's perception.

As for the decision of why this municipality, this is explained in the introduction by representing the state of Queretaro as one of the most productive in the country, but where the population of Pinal de Amoles is located is the one that registers the highest poverty conditions in the state of Queretaro, Mexico.

Reviewer # 1, Concern # 9: Figure 2 could be interesting but needs to be explained as well as Figure 3. 269 needs a number page.

Author response: We have attended to this comment. Thank you very much!

Author action: This paragraph clarifies that it refers to the chord diagram of the relationships between subsystems. It is further explained in the introduction.

Reviewer # 1, Concern # 10: Check spacing 286-297. Why the "computational simulation" is a new section? 

Author response: Excellent observation!

Author action: We have considered that this paragraph represents the continuity of the text and not another section, therefore, it was eliminated.

Reviewer # 1, Concern # 11: RESULTS: there is already confusion on materials and results. Please check. Figure 4 could be more useful in the section 2.

Author response: ¡We consider this an appropriate comment, thank you!

Author action: We have considered integrating Figure 4 into Section 2 and believe it is a correct decision.

Reviewer # 1, Concern # 12: DISCUSSION AND CONCLUSION: please use the same order criteria to reframe the texts.

Author response: ¡This aspect improves the order of ideas, thank you very much!

Author action: The criteria for presenting the texts in both sections have been corrected.

Reviewer # 1, Concern # 13: Check editing of bullet points, typos, spacing, and repetitions, and use adequate adverbs

Author response: Thanks for your suggestion.

Author action: The bullets and adverbs used have been revised.

Reviewer # 1, Concern # 14:  I would suggest to use always numbers for headings and sub headings to facilitate the work of the reader and give always a reference to the main section (3; 3.1; 3.1.1).

Author response: We appreciate your valuable suggestion in this regard.

Author action: The sections have been duly ordered according to their follow-up.

Reviewer 2 Report

This paper is now much improved, so it can be accepted for publication.

Author Response

Reply to the reviewer’s comments

We would like to thank the reviewers for their efforts and interest in our work. Your comments will improve the quality of this manuscript and give us the opportunity to clarify these comments. We will respond to all the comments raised by the reviewers.

We have carefully reviewed and attended to all comments raised by the reviewers, as listed below, highlighted in the manuscript.

Reviewer # 2. We really appreciate your comments and suggestions.

Reviewer 3 Report

The manuscript has been sufficiently improved based on the given comments. It has been developed theoretically. The methodological part of the article has also been developed. It has now clearly stated contribution in the article. I can see that the internal validity of the revised manuscript has also been increased. From my point of view, the article is ready for publication. 

Author Response

Reply to the reviewer 3 comments

We would like to thank the reviewers for their efforts and interest in our work. Your comments will improve the quality of this manuscript and give us the opportunity to clarify these comments. We will respond to all the comments raised by the reviewers.

We have carefully reviewed and attended to all comments raised by the reviewers, as listed below, highlighted in the manuscript.

Reviewer # 3. We really appreciate your comments and suggestions.
